

# Hamiltonian truncation effective theory

Timothy Cohen[1], Kara Farnsworth[2], Rachel Houtz[3] and Markus A. Luty[4]

**1** Institute for Fundamental Science, University of Oregon, Eugene, Oregon 97403, USA
**2** CERCA, Department of Physics, Case Western Reserve University,
Cleveland, Ohio 44106, USA
**3** Institute for Particle Physics Phenomenology, Department of Physics,
Durham University, Durham DH1 3LE, U.K.
**4** Center for Quantum Mathematics and Physics (QMAP), University of California,
Davis, California 95616, USA

## Abstract

Hamiltonian truncation is a non-perturbative numerical method for calculating observables of a quantum field theory. The starting point for this method is to truncate the interacting Hamiltonian to a finite-dimensional space of states spanned by the eigenvectors of the free Hamiltonian $H_0$ with eigenvalues below some energy cutoff $E_{\max}$. In this work, we show how to treat Hamiltonian truncation systematically using effective field theory methodology. We define the finite-dimensional effective Hamiltonian by integrating out the states above $E_{\max}$. The effective Hamiltonian can be computed by matching a transition amplitude to the full theory, and gives corrections order by order as an expansion in powers of $1/E_{\max}$. The effective Hamiltonian is non-local, with the non-locality controlled in an expansion in powers of $H_0/E_{\max}$. The effective Hamiltonian is also non-Hermitian, and we discuss whether this is a necessary feature or an artifact of our definition. We apply our formalism to 2D $\lambda \phi^4$ theory, and compute the the leading $1/E_{\max}^2$ corrections to the effective Hamiltonian. We show that these corrections non-trivially satisfy the crucial property of separation of scales. Numerical diagonalization of the effective Hamiltonian gives residual errors of order $1/E_{\max}^3$, as expected by our power counting. We also present the power counting for 3D $\lambda \phi^4$ theory and perform calculations that demonstrate the separation of scales in this theory.

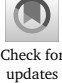

# 1 Introduction

Numerical methods for studying strongly interacting quantum field theories and quantum many-body systems are an important component of the modern physics toolkit. Two of the most commonly used methods are lattice Monte Carlo (*e.g.* lattice gauge theory) and the density matrix renormalization group (used mainly in condensed matter physics). The focus of this paper is on a less-frequently used approach known as Hamiltonian truncation, a numerical method that diagonalizes the Hamiltonian projected onto a finite-dimensional subspace of the full Hilbert space. The method goes back to the earliest days of quantum mechanics, where it

is known as the Rayleigh-Ritz variational method. Its first use in quantum field theory appears to be Ref. [1]. The method was applied to renormalization group flows between 2D conformal field theories in [2,3], where it was called the 'truncated conformal space approach.' This work demonstrated the effectiveness of the method applied to 2D quantum field theories, and led to many applications both in elementary particle theory and condensed matter theory (see [4] for a review). More recently, there has been a revival of interest in Hamiltonian truncation in the quantum field theory literature following the pioneering works Refs. [5–7]. See Refs. [8–24] for examples of subsequent developments and applications.

In this paper, we study a version of Hamiltonian truncation where the finite-dimensional Hilbert space is defined using an energy cutoff, since in this case we expect to be able to apply the ideas and techniques of low-energy effective field theory [25]. Specifically, the full Hamiltonian is written

$$H = H_0 + V, \tag{1}$$

where $H_0$ is a free Hamiltonian that can be diagonalized exactly. The finite-dimensional Hilbert space $\mathcal{H}_{\text{eff}}$ is defined to be linear combinations of $H_0$ eigenstates

$$H_0|E_i\rangle = E_i|E_i\rangle, \tag{2}$$

with $E_i \leq E_{\text{max}}$. We are interested in theories where the interactions in $V$ are weak in the UV, for example theories with relevant couplings. For such theories, we will show how to systematically construct the effective Hamiltonian as an expansion in $1/E_{\text{max}}$ in perturbation theory. We expect that physical quantities sensitive to energies well below $E_{\text{max}}$ can be approximated by such an effective Hamiltonian.

A fundamental limitation of Hamiltonian truncation is that the number of states in $\mathcal{H}_{\text{eff}}$ grows exponentially with $E_{\text{max}}$, while the accuracy is expected to decrease as a power of $1/E_{\text{max}}$. Since the computational resources scale with the number of states, the accuracy only improves logarithmically with computational resources, at least for conventional computational methods.[1] Despite this limitation, interesting levels of accuracy have been obtained using Hamiltonian truncation in low-dimensional systems (reviewed in [4]). Furthermore, this method has the potential to perform calculations in theories that are not easily treated with lattice methods, for example theories with chiral fermions [20] or theories with sign problems [8,10]. This strongly motivates further study of Hamiltonian truncation to determine its ultimate potential. Our focus in the present paper is improving the convergence of the method as a function of the cutoff $E_{\text{max}}$, a problem that has already been studied in [9,13–15,26]. We will compare our method with these works in §8.

In this paper, we develop a systematic approach to Hamiltonian truncation using the methodology of effective field theory. We call the resulting formalism Hamiltonian Truncation Effective Theory (HTET). The essential idea is that the truncation parameter $E_{\text{max}}$ is treated as a UV cutoff, and the finite-dimensional effective Hamiltonian is defined by matching to the full theory. The resulting theory has many of the expected features of more conventional effective field theories, but also has some unique features that result from the nature of the cutoff:

- We define the effective Hamiltonian by matching a transition amplitude that is well-defined in finite volume order by order in powers of $V$. The matching can be carried out using a systematic diagrammatic expansion similar to time-ordered perturbation theory.

---

[1]Quantum computers can efficiently store an exponentially large Hilbert space with linear resources (qubits). The development of a complete quantum algorithm for Hamiltonian truncation is an interesting problem for future work.

- The effective Hamiltonian is non-local. This arises because the cutoff of the effective theory is non-local: $E_{\mathrm{max}}$ is the maximum value of the *total* energy (defined by $H_0$) of the system, which gets contributions from all excitations regardless of how far apart they are.

- The effective Hamiltonian is non-Hermitian. Because time evolution with the full Hamiltonian mixes states above and below the cutoff, there is no physical reason to expect the effective Hamiltonian to be Hermitian. We discuss whether the non-Hermiticity is a necessary feature, or an artifact of our definitions. We give some arguments that non-Hermiticity is necessary to maintain desirable properties of the effective Hamiltonian, but we leave a full discussion for future work.

- We propose a power counting to all orders in the $1/E_{\mathrm{max}}$ expansion in which the non-Hermiticity and non-locality are controlled by an expansion in powers of $H_0/E_{\mathrm{max}}$.

- Our formalism can be applied without modification to theories with non-trivial UV divergences. The renormalized fundamental theory gives predictions that are finite and independent of $E_{\mathrm{max}}$, so matching is expected to give an effective Hamiltonian whose predictions are finite and independent of $E_{\mathrm{max}}$.

The crucial property of any effective theory is that it factorizes physical effects associated with different scales. This means that the effective Hamiltonian depends on the properties of states above the cutoff, and parameterizes the effects below the cutoff. In perturbative matching calculations such as the one developed in this paper, the separation of scales has a precise meaning: corrections to the effective Hamiltonian are given by sums over states that are dominated by states near the effective theory cutoff. In particular, this implies that the effective theory matching corrections are insensitive to IR modifications of the theory, such as masses and compact spatial dimensions, as long as the mass scale of these modifications is small compared to the cutoff scale. The calculations performed in this paper for $\lambda\phi^4$ theory in 2D and 3D give a nontrivial demonstration of this property, involving 2- and 3- loop diagrams with overlapping UV/IR dominated regions that cancel only with the correct operator definition and renormalization scheme.

To check that this formalism actually improves the numerical convergence as expected, we perform numerical calculations for 2D $\lambda\phi^4$ theory. We find that the size of the numerical error is compatible with the theoretically predicted scaling with powers of $1/E_{\mathrm{max}}$, namely $O(1/E_{\mathrm{max}}^2)$ for the $O(\lambda)$ effective Hamiltonian and $O(1/E_{\mathrm{max}}^3)$ at $O(\lambda^2)$. At this order, the effective Hamiltonian is local and Hermitian. Going to higher orders is simply a matter of computing additional diagrams, and we show by explicit computation that there are non-local and non-Hermitian $1/E_{\mathrm{max}}^3$ corrections to the effective Hamiltonian. In future work, we plan to extend our calculations in 2D $\lambda\phi^4$ theory to higher order, and to perform numerical calculations for 3D $\lambda\phi^4$ theory. This will check that our formalism works when the effective Hamiltonian is non-local, and in theories with UV divergences.

The rest of this paper is organized as follows. We derive a general formalism for matching onto the effective Hamiltonian in §2. In §3 we derive a set of diagrammatic rules to compute the transition amplitude that is used to perform the matching, using 2D $\lambda\phi^4$ theory as an example. The renormalization and matching for 2D $\lambda\phi^4$ theory are presented in §4 and §5 respectively. We discuss the power counting of the effective theory in §6, and give an explicit example of non-local and non-Hermitian terms that appear at higher orders in the expansion. Our numerical results for this theory are presented in §7. In §8, we compare our formalism and results to previous results in the literature. We conclude in §9 with a discussion of future directions. In Appendix A, we present some calculations for 3D $\lambda\phi^4$ theory to illustrate the application of our methods to theories with non-trivial UV divergences. In Appendix B, we

consider the possibility of defining a Hermitian effective Hamiltonian by a similarity transformation.

## 2 Effective Hamiltonian from Matching

In this section, we define the effective Hamiltonian by matching to the predictions of the full theory order by order in an expansion in powers of $V$.[2] This definition is the basis for the systematic computation the effective Hamiltonian in an expansion in $1/E_{\max}$, which we present in §6. We make the separation of the full Hamiltonian into the free and interacting parts defined in Eq. (1), and assume that the spectrum of $H_0$ (and $H$) is discrete, with

$$H_0|i\rangle = E_i|i\rangle, \qquad i = 0, 1, 2, \dots . \tag{3}$$

We denote the full Hilbert space by $\mathcal{H}$, and the finite-dimensional subspace spanned by the states $|i\rangle$ with $E_i \leq E_{\max}$ by $\mathcal{H}_{\text{eff}}$. Our goal is to define an effective Hamiltonian $H_{\text{eff}}$ acting on the effective Hilbert space $\mathcal{H}_{\text{eff}}$, so that the low-lying eigenvalues of $H_{\text{eff}}$ approximate those of $H$.

The simplest approximation for the effective Hamiltonian is

$$\langle f|H_{\text{eff}}|i\rangle \simeq \langle f|H|i\rangle, \tag{4}$$

for $|i\rangle, |f\rangle \in \mathcal{H}_{\text{eff}}$. That is, $H_{\text{eff}}$ is the restriction of $H$ to the low-energy subspace $\mathcal{H}_{\text{eff}}$. The goal of our formalism is to systematically improve this approximation for theories in which the interaction $V$ can be treated as a perturbation in the UV. For 2D $\lambda\phi^4$ theory, Eq. (4) is a good starting approximation, and we will focus mainly on that case in this paper. For theories with non-trivial UV divergences, one must go to higher orders in the matching to obtain a good starting approximation. We will discuss the example of 3D $\lambda\phi^4$ theory in Appendix A.

The purpose of the effective theory approach is to systematically improve the approximation Eq. (4) by including the effects of the states above the cutoff $E_{\max}$. In theories that are weakly coupled in the UV, we expect that the effects of the states above $E_{\max}$ can be computed order by order in powers of $V$. In the matching approach used here, this is done by matching a physical quantity in the fundamental and effective theory order by order in powers of $V$. This means that we choose some observable that can be computed in both the fundamental and effective theory, and define the effective Hamiltonian by requiring that the physical quantities agree. An obvious physical quantity to match in Hamiltonian truncation is the spectrum of energy eigenvalues. However, we will now show that this is not sufficient to completely define the effective Hamiltonian.

### 2.1 Matching the Spectrum

For simplicity, we assume that the spectrum of $H_0$ is non-degenerate. In perturbation theory, there is a one-to-one correspondence between eigenvalues of $H$ and $H_0$:

$$H|i\rangle = \mathcal{E}_i|i\rangle, \tag{5}$$

where

$$\mathcal{E}_i = E_i + \mathcal{E}_{1i} + \mathcal{E}_{2i} + \cdots, \qquad \text{with} \qquad \mathcal{E}_{ni} = O(V^n). \tag{6}$$

---

[2]Our definition differs from the 'exact effective Hamiltonian' of Refs. [5,6], which is a function of the energy eigenvalue that is being computed. See §8.3.

Here $|i\rangle \in \mathcal{H}_{\text{eff}}$ and we have already set $\mathcal{E}_{0i} = E_i$ from Eq. (3) since we are working in the basis of the unperturbed Hamiltonian $H_0$. We write

$$H_{\text{eff}} = H_0 + H_1 + H_2 + \cdots, \qquad H_n = O(V^n) \tag{7}$$

and use standard Rayleigh-Schrödinger perturbation theory compute the eigenvalues as an expansion in powers of $V$ in the effective theory:

$$\mathcal{E}_{1i} = \langle i|H_1|i\rangle, \tag{8a}$$

$$\mathcal{E}_{2i} = \sum_{\alpha \neq i}^{<} \frac{\left|\langle i|H_1|\alpha\rangle\right|^2}{E_{i\alpha}} + \langle i|H_2|i\rangle, \tag{8b}$$

$$\mathcal{E}_{3i} = \sum_{\alpha,\beta \neq i}^{<} \frac{\langle i|H_1|\alpha\rangle\langle\alpha|H_1|\beta\rangle\langle\beta|H_1|i\rangle}{E_{i\alpha}E_{i\beta}}$$
$$+ \sum_{\alpha \neq i}^{<} \frac{1}{E_{i\alpha}}\bigg[ \langle i|H_1|\alpha\rangle\langle\alpha|H_2|i\rangle + \langle i|H_2|\alpha\rangle\langle\alpha|H_1|i\rangle$$
$$- \frac{\langle i|H_1|\alpha\rangle\langle\alpha|H_1|i\rangle\langle i|H_1|i\rangle}{E_{i\alpha}}\bigg] + \langle i|H_3|i\rangle, \tag{8c}$$

where we write $E_{i\alpha} = E_i - E_\alpha$ and

$$\sum_{\alpha}^{<} = \sum_{E_\alpha \leq E_{\max}} \qquad \text{and} \qquad \sum_{\alpha}^{>} = \sum_{E_\alpha > E_{\max}}. \tag{9}$$

These restricted sums appear because $H_{\text{eff}}$ is an operator on $\mathcal{H}_{\text{eff}}$, so that the intermediate states in the effective theory must be restricted to the low-energy subspace.

It is straightforward to see that Eqs. (8) do not uniquely define $H_{\text{eff}}$ order by order in the expansion in $V$. We assume that $\mathcal{E}_{ni}$ have been computed in the fundamental theory, and we are using Eqs. (8) to determine the corrections $H_n$ in the effective theory. At first order (see Eq. (8a)), the matching only determines the diagonal elements of $H_1$. At order $V^n$ with $n > 1$, the matching depends on the off-diagonal components of $H_m$ with $m < n$, as well as the diagonal elements of $H_n$. We therefore need additional relations to fully define the effective Hamiltonian. In the following section, we show that matching a different quantity, related to the $S$-matrix, completely defines $H_{\text{eff}}$ order by order in perturbation theory.

## 2.2 The Transition Matrix

We showed above that matching the stationary states does not uniquely define the effective Hamiltonian. We need a complete definition for the effective Hamiltonian as a starting point for systematic expansion. In this subsection we will show that the effective Hamiltonian can be defined order by order in powers of $V$ by matching a transition amplitude. It is natural to guess that the effective Hamiltonian can be defined in this way. In effective field theories in infinite volume, the effective Hamiltonian can be defined by matching the $S$-matrix, the unitary time evolution operator between asymptotic scattering states. The $S$-matrix is not well-defined in finite volume, but we will show that we can define the effective Hamiltonian by matching a similar observable.

We start with an initial state at $t_i = 0$. We then turn off the interactions adiabatically for $t > 0$ by making the replacement

$$V \to Ve^{-\epsilon t}, \tag{10}$$

where $\epsilon > 0$ is an infinitesimal regulatory parameter with units of energy. We then evolve the state to $t_f \to +\infty$, and compute its overlap with an eigenstate of $H_0$:

$$\lim_{t_f \to \infty} \langle f | e^{-iH_{\text{eff}} t_f} | i \rangle \overset{?}{=} \lim_{\epsilon \to 0^+} \langle f | \text{T} \exp \left\{ -i \int_0^\infty dt \left( H_0 + V e^{-\epsilon t} \right) \right\} | i \rangle, \tag{11}$$

where T is the time ordering operator. This is still not quite what we want, because it involves an ill-defined infinite phase in the limit $t_f \to \infty$. However, we can factor out this phase by working in the interaction picture, defined in terms of Schrödinger picture by

$$|\Psi(t)\rangle_{\text{IP}} = e^{iH_0 t} |\Psi(t)\rangle_{\text{SP}}, \qquad \mathcal{O}_{\text{IP}}(t) = e^{iH_0 t} \mathcal{O}_{\text{SP}} e^{-iH_0 t}. \tag{12}$$

The interaction picture time evolution operator is

$$U_{\text{IP}}(t_f, t_i) = \text{T} \exp \left\{ -i \int_{t_i}^{t_f} dt \, V_{\text{IP}}(t) \right\}, \qquad \text{with} \qquad V_{\text{IP}}(t) = e^{iH_0 t} V e^{-\epsilon t} e^{-iH_0 t}. \tag{13}$$

We then define the operator

$$\langle f | \Sigma(\epsilon) | i \rangle \equiv \lim_{t_f \to \infty} \langle f | U_{\text{IP}}(t_f, 0) | i \rangle, \tag{14}$$

where we emphasize that $\epsilon \neq 0$ in the definition of $\Sigma$.

We now work out the perturbative expansion of $\Sigma$. To do this, we note that the time evolution operator written in Eq. (13) obeys

$$\frac{\partial}{\partial t_i} U_{\text{IP}}(t_f, t_i) = i U_{\text{IP}}(t_f, t_i) V_{\text{IP}}(t_i), \tag{15}$$

with boundary condition $U_{\text{IP}}(t_f, t_f) = \mathbb{1}$. The solution is

$$U_{\text{IP}}(t_f, t_i) = \mathbb{1} - i \int_{t_i}^{t_f} dt \, U_{\text{IP}}(t_f, t) V_{\text{IP}}(t). \tag{16}$$

This all-orders relation can be expanded iteratively in powers of $V$. Applying this to Eq. (14), the leading terms are

$$\langle f | \Sigma | i \rangle = \delta_{fi} + \frac{\langle f | V | i \rangle}{E_{fi} + i\epsilon} + \sum_\alpha \frac{\langle f | V | \alpha \rangle \langle \alpha | V | i \rangle}{(E_{fi} + i\epsilon)(E_{f\alpha} + i\epsilon)} + O(V^3), \tag{17}$$

where $E_{f\alpha} \equiv E_f - E_\alpha$, as before. This expression is not symmetric under the exchange $i \leftrightarrow f$ due to the fundamental time asymmetry built into the definition of $\Sigma$ given in Eq. (14). Note that the right-hand side of Eq. (17) is actually not well-defined because some of the energy denominators vanish for $\epsilon = 0$.[3] However, we will see below that these singular terms cancel

---

[3] We note that $\Sigma$ can be made well-defined in the $\epsilon \to 0$ limit if one chooses to define $V$ so that it does not contain any diagonal terms when expressed in the $H_0$ eigenbasis. That is, we make the replacements

$$H_0 \to H_0 + V_{\text{diag}}, \qquad V \to V - V_{\text{diag}}, \tag{18}$$

where

$$V_{\text{diag}} = \sum_i |i\rangle \langle i | V | i \rangle \langle i |, \tag{19}$$

is the diagonal part of $V$. As in perturbation theory for eigenvectors, this ensures that the perturbation of the states are perpendicular to the unperturbed states. The result for the effective Hamiltonian (Eq. (25) below) is invariant under this shift. However, this approach is not convenient for the diagrammatic expansion discussed in §3 below, and so we will not pursue it further.

in the matching. We can therefore treat $\epsilon$ as an IR regulator, taking the limit $\epsilon \to 0$ after performing the matching calculation. The fact that the matching is insensitive to the IR details of the theory is an important feature of the effective theory approach that will be discussed further in §5 below.

It is convenient to remove the $E_{fi}$ energy denominator that is common to all terms by defining the $T$-matrix:

$$\langle f|\Sigma|i \rangle = \delta_{fi} + \frac{\langle f|T|i \rangle}{E_{fi} + i\epsilon} . \tag{20}$$

By analogy with the $S$-matrix, we will refer to $T$ as the 'transition matrix.' The first few terms in the perturbative expansion of $T$ are then

$$\langle f|T|i \rangle = \langle f|V|i \rangle + \sum_{\alpha} \frac{\langle f|V|\alpha \rangle\langle \alpha|V|i \rangle}{E_{f\alpha} + i\epsilon} + \sum_{\alpha,\beta} \frac{\langle f|V|\alpha \rangle\langle \alpha|V|\beta \rangle\langle \beta|V|i \rangle}{(E_{f\alpha} + i\epsilon)(E_{f\beta} + i\epsilon)} + O(V^4), \tag{21}$$

or in other words, the $O(V^n)$ contribution to $\langle f|T|i \rangle$ is given by

$$T_n = \sum_{\alpha_1,\dots,\alpha_{n-1}} \frac{\langle f|V|\alpha_1 \rangle\langle \alpha_1|V|\alpha_2 \rangle \cdots \langle \alpha_{n-1}|V|i \rangle}{(E_{f\alpha_1} + i\epsilon) \cdots (E_{f\alpha_{n-1}} + i\epsilon)} . \tag{22}$$

Note that the sum includes terms where the energy denominators $E_{f\alpha}$ vanish, and therefore the transition matrix diverges as $\epsilon \to 0$. We can think of $\epsilon$ as an IR regulator, so these terms are IR divergent. However, we will show that these IR divergences cancel in the matching, and the effective Hamiltonian is completely defined by matching $T$ order by order in powers of $V$.

## 2.3 Matching the Transition Matrix

To match, we must compute the transition matrix $T$ in the effective theory. In terms of

$$V_{\text{eff}} = H_1 + H_2 + \cdots , \tag{23}$$

the interaction picture time evolution operator is given by

$$U_{\text{eff, IP}}(t_f, t_i) = T \exp\left(-i \int_{t_i}^{t_f} dt \, V_{\text{eff, IP}}(t)\right), \quad \text{with} \quad V_{\text{eff, IP}}(t) = e^{iH_0 t} V_{\text{eff}} e^{-\epsilon t} e^{-iH_0 t} . \tag{24}$$

Sums over states in the effective theory are restricted to satisfy $E \le E_{\max}$. Matching the matrix elements of the transition matrix to the fundamental theory, we then obtain our matching conditions:

$$\langle f|H_1|i \rangle_{\text{eff}} = \langle f|V|i \rangle , \tag{25a}$$

$$\langle f|H_2|i \rangle_{\text{eff}} = \sum_{\alpha}^{>} \frac{\langle f|V|\alpha \rangle\langle \alpha|V|i \rangle}{E_{f\alpha}} , \tag{25b}$$

$$\langle f|H_3|i \rangle_{\text{eff}} = \sum_{\alpha,\beta}^{>} \frac{\langle f|V|\alpha \rangle\langle \alpha|V|\beta \rangle\langle \beta|V|i \rangle}{E_{f\alpha}E_{f\beta}} - \sum_{\alpha}^{<}\sum_{\beta}^{>} \frac{\langle f|V|\alpha \rangle\langle \alpha|V|\beta \rangle\langle \beta|V|i \rangle}{E_{\alpha\beta}E_{f\beta}} . \tag{25c}$$

The matrix elements of the effective Hamiltonian are written with a subscript 'eff' to remind us that the matrix elements are evaluated in the finite-dimensional Hilbert space $\mathcal{H}_{\text{eff}}$. Note that for $E_i, E_f \ll E_{\max}$ the energy denominators in Eq. (25) are of order $E_{\max}$ or larger. This

reflects the fact that the matching calculation is insensitive to the IR details of the theory, and is important for the separation of scales, as we discuss below. It also means that we can take the limit $\epsilon \to 0$ in the matching, so the IR divergences in the transition matrix $T$ do not affect the matching.

$H_{\text{eff}}$ as given in Eqs. (25) defines the effective Hamiltonian order by order in powers of $V$. We have verified to $O(V^4)$ that the eigenvalues computed using $H_{\text{eff}}$ agree with those obtained from directly matching the eigenvalues Eq. (8). A peculiar feature of the effective Hamiltonian defined here is that it is not Hermitian. Note that time evolution with the full Hamiltonian $H$ mixes states in the low-energy subspace $\mathcal{H}_{\text{eff}}$ with states that are not in $\mathcal{H}_{\text{eff}}$. Therefore, there is no reason *a priori* that the effective Hamiltonian must be Hermitian. However, it is interesting to explore if there exists an alternative definition of the effective Hamiltonian that is Hermitian. Beyond the conceptual implications, this is motivated by the fact that numerical algorithms for diagonalizing Hermitian matrices are more efficient than for non-Hermitian matrices.

It is not obvious that $T$ is the 'correct' observable to match. Is it possible to define a Hermitian effective Hamiltonian that has the desired properties, such as separation of scales? We do not have a complete answer, but we have considered a few simple possibilities, and find that they do not work. For example, we can define a Hermitian effective Hamiltonian by matching $T + T^\dagger$. At $O(V^2)$ this gives an effective Hamiltonian that is related to the Schrieffer-Wolf effective Hamiltonian [27]), but at $O(V^3)$ the effective Hamiltonian defined in this way diverges in the limit $\epsilon \to 0$.[4] This is an IR divergence, signaling a failure of separation of scales.

Another possibility is to define a new effective Hamiltonian from our $H_{\text{eff}}$ by a similarity transformation $H'_{\text{eff}} = G H_{\text{eff}} G^{-1}$. This does not change the spectrum, so $H'_{\text{eff}}$ is a suitable effective Hamiltonian. If $G$ is not unitary, it may be possible to obtain a Hermitian effective Hamiltonian in this way. In Appendix B, we present a choice of $G$ that removes some but not all of the leading non-Hermitian terms in 2D $\lambda \phi^4$ theory.

It would be be interesting to investigate alternative definitions of the effective Hamiltonian, but we leave this for future work.

# 3 Diagrammatic Rules

It is very useful to have a diagrammatic expansion for the transition matrix $T$ to perform the matching. In this section, we derive such a set of diagrammatic rules. This requires specifying

---

[4]At $O(V^2)$, this gives the Hermitian average of our effective Hamiltonian

$$\langle f | H_2 | i \rangle = \frac{1}{2} \sum_\alpha^{>} \langle f | V | \alpha \rangle \langle \alpha | V | i \rangle \left( \frac{1}{E_{f\alpha}} + \frac{1}{E_{i\alpha}} \right), \tag{26}$$

but at $O(V^3)$ we obtain

$$\begin{aligned}
\langle f | H_3 | i \rangle = {} & \frac{1}{2} \sum_{\alpha,\beta}^{>} \langle f | V | \alpha \rangle \langle \alpha | V | \beta \rangle \langle \beta | V | i \rangle \left( \frac{1}{E_{f\alpha} E_{f\beta}} + \frac{1}{E_{i\alpha} E_{i\beta}} \right) \\
& + \frac{1}{4} \sum_\alpha^{>} \sum_\beta^{<} \langle f | V | \alpha \rangle \langle \alpha | V | \beta \rangle \langle \beta | V | i \rangle \frac{E_{i\alpha}^2 + E_{i\beta} E_{f\alpha} + E_{f\alpha} E_{\alpha\beta}}{E_{i\beta} E_{i\alpha} E_{f\alpha} E_{\alpha\beta}} \\
& - \frac{1}{4} \sum_\alpha^{<} \sum_\beta^{>} \langle f | V | \alpha \rangle \langle \alpha | V | \beta \rangle \langle \beta | V | i \rangle \frac{E_{f\beta}^2 + E_{f\alpha} E_{i\beta} - E_{i\beta} E_{\alpha\beta}}{E_{f\alpha} E_{f\beta} E_{i\beta} E_{\alpha\beta}} \, . \tag{27}
\end{aligned}$$

We have omitted $i\epsilon$ terms for brevity. This contains IR divergent terms where the energy denominators vanish, for example the terms with $\alpha = f$ in the last line.

a model, and we use 2D $\lambda\phi^4$ theory as an example. This diagrammatic expansion is similar to 'old-fashioned perturbation theory' for the $S$-matrix.

Not only will these rules serve as a useful calculational tool, but they also provide insight into the properties of the effective Hamiltonian. For example, the apparent non-locality of the effective Hamiltonian that appears at subleading order in the $1/E_{\max}$ expansion will have a simple diagrammatic interpretation. Additionally, the diagrammatic expansion also illuminates the UV divergence structure of the theory, making the interplay between matching and renormalization completely transparent.

We consider the Lagrangian density

$$\mathcal{L} = \frac{1}{2}(\partial\phi)^2 - \frac{1}{2}m^2\phi^2 - \frac{\lambda}{4!}\phi^4. \tag{28}$$

The mass dimension of the fields and couplings in 2D are

$$[\phi] = 0, \qquad [\lambda] = [m^2] = 2. \tag{29}$$

The $\phi^4$ coupling is relevant, meaning that this interaction is weak in the UV and strong in the IR.

We quantize the theory on a spatial circle of radius $R$:

$$\phi(x + 2\pi R, t) = \phi(x, t). \tag{30}$$

We can therefore expand the fields in a discrete set of momentum modes:

$$\phi(x, t) = \frac{1}{\sqrt{2\pi R}} \sum_{k \in \mathbb{Z}} e^{ikx/R} \phi_k(t). \tag{31}$$

The fact that $\phi$ is Hermitian implies

$$\phi_k^\dagger = \phi_{-k}. \tag{32}$$

To simplify the calculations, it is useful to define the vertices to be normal-ordered operators. This means that we choose a 'quantization mass' $m_Q$ and write the Hamiltonian in terms of creation and annihilation operators for Fock states with mass $m_Q$:

$$H_0 = \sum_k \omega_k a_k^\dagger a_k, \qquad \text{with} \qquad \omega_k = \sqrt{(k/R)^2 + m_Q^2}, \tag{33}$$

where creation and annihilation operators obey the standard canonical commutation relations

$$\left[a_k, a_{k'}^\dagger\right] = \delta_{kk'}. \tag{34}$$

The interaction term is then given by

$$V = \int dx \left[\frac{1}{2}m_V^2 :\phi^2: + \frac{\lambda}{4!} :\phi^4:\right], \tag{35}$$

where $m_V$ is an 'interaction mass' that contributes to $V$ and $:\mathcal{O}:$ denotes the normal ordering of the operator $\mathcal{O}$ with respect to the creation and annihilation operators defined above. For the purpose of deriving the diagrammatic rules, we treat the parameters $m_Q$ and $m_V$ as finite quantities; the renormalization of the theory is discussed in §4 below. We write

$$\phi_k = \phi_k^{(+)} + \phi_{-k}^{(-)}, \tag{36}$$

where

$$\phi_k^{(+)} = \frac{1}{\sqrt{2\omega_k}}a_k, \qquad \phi_k^{(-)} = \frac{1}{\sqrt{2\omega_k}}a_k^\dagger. \tag{37}$$

(Note that Eq. (36) satisfies Eq. (32).)

We then obtain the diagrammatic rules using Wick's theorem. The products of $V$ that appear in $T$ are not time-ordered (see Eq. (22)), so the relevant contraction is defined by

$$\phi_1\phi_2 = {:}\phi_1\phi_2{:} + \underbrace{\phi_1\phi_2}, \tag{38}$$

where

$$\underbrace{\phi_k\phi_{k'}} = \frac{\delta_{kk'}}{2\omega_k}. \tag{39}$$

Because we are writing $V$ in terms of normal-ordered operators, the version of Wick's theorem we are using is

$$\left({:}\mathcal{O}_1{:}\right)\cdots\left({:}\mathcal{O}_n{:}\right) = {:}\left(\mathcal{O}_1\cdots\mathcal{O}_n\right){:} + \text{contractions}, \tag{40}$$

where contractions between fields in the same operator are omitted. For example,

$${:}\phi_1^2{:}{:}\phi_2^2{:} = {:}\phi_1^2\phi_2^2{:} + 4\underbrace{\phi_1\phi_2}{:}\phi_1\phi_2{:} + 2\left(\underbrace{\phi_1\phi_2}\right)^2. \tag{41}$$

The normal-ordered operators that appear in Wick's theorem consist of a sum of terms with powers of $\phi^{(+)}$ acting on initial states to the right, and powers of $\phi^{(-)}$ acting on final states to the left. Each monomial in $\phi^{(\pm)}$ is represented by a sum of diagrams. As usual, we denote the powers of $V$ by vertices, and the Wick contractions by lines connecting to other vertices or external states. The vertices in the diagram are ordered from left to right in the same order that the insertions of $V$ appear in Eq. (22). The diagrammatic rules for $T$ at order $V^n$ can be summarized as follows:

- Draw all possible diagrams with $n$ ordered vertices. Each line is either connected to a different vertex, to the initial state on the right, or the final state on the left. Disconnected diagrams must be included.

- Assign an independent mode number $k$ to each internal and external line. Sum over the internal momenta.

- The rules for the vertices are

$$k_3 \diagdown\!\!\!\!\diagup k_1 \atop k_4 \diagup\!\!\!\!\diagdown k_2 = \frac{\lambda}{2\pi R}\delta_{k_1+\cdots+k_4}, \qquad k_2 \longrightarrow\!\!\bullet\!\!\longrightarrow k_1 = m_V^2\delta_{k_1 k_2}, \tag{42}$$

where the momenta are taken to all flow into the vertex.

- Each internal line is associated with a factor of

$$\longrightarrow = \frac{1}{2\omega_k}. \tag{43}$$

- A diagram with $n$ lines going to the initial state and $m$ lines going to the final state contains the factor

$$\langle f|\phi_{k_{n+m}}^{(-)}\cdots\phi_{k_{n+1}}^{(-)}\phi_{k_n}^{(+)}\cdots\phi_{k_1}^{(+)}|i\rangle. \tag{44}$$

- Each vertex is associated with an energy denominator, given by

$$\frac{1}{E_{f\alpha} + i\epsilon} = \frac{1}{E_f - E_\alpha + i\epsilon},$$ (45)

where $E_f$ is the energy of the final state, and $E_\alpha$ is the energy of the state directly to the right of the vertex. The energy denominator associated with the rightmost vertex (which would give a factor of $1/E_{fi}$) is omitted (see Eq. (20)). The initial and final states in general contain particles that do not participate in the interaction, but these do not contribute to the energy differences.

- Multiply by the symmetry factor

$$S = \left(\frac{1}{4!}\right)^{n_4} \left(\frac{1}{2}\right)^{n_2} C,$$ (46)

where $n_4$ ($n_2$) is the number of $\phi^4$ ($\phi^2$) vertices, and $C$ is the number of Wick contractions that give the same diagram. To count the contractions, the initial and final state particles should be treated as identical, but the initial state particles can be distinguished from final state particles. In terms of operators, $C$ is the coefficient of the operator that appears in front of the matrix element of the form in Eq. (44) when using Wick's theorem.[5]

We give some examples to illustrate these rules:

$$\phantom{x}_{4}^{3} \!\!\!\gtrless\!\!\!\begin{smallmatrix}5\\6\end{smallmatrix}\!\!\!\lessgtr\!\!\!\phantom{x}_{2}^{1} = \frac{1}{8}\left(\frac{\lambda}{2\pi R}\right)^2 \sum_{1,\ldots,6} \delta_{12,56}\delta_{34,56}\langle f|\phi_4^{(-)}\phi_3^{(-)}\phi_2^{(+)}\phi_1^{(+)}|i\rangle$$
$$\times \frac{1}{2\omega_5}\frac{1}{2\omega_6}\frac{1}{\omega_3+\omega_4-\omega_5-\omega_6+i\epsilon},$$ (48a)

$$\phantom{x}_{4}^{3} \!\!\!\gtrless\!\!\!\begin{smallmatrix}5\,6\end{smallmatrix}\!\!\!\gtrless\!\!\!\phantom{x}_{2}^{1} = \frac{1}{8}\left(\frac{\lambda}{2\pi R}\right)^2 \sum_{1,\ldots,6} \delta_{12,56}\delta_{34,56}\langle f|\phi_4^{(-)}\phi_3^{(-)}\phi_2^{(+)}\phi_1^{(+)}|i\rangle$$
$$\times \frac{1}{2\omega_5}\frac{1}{2\omega_6}\frac{1}{-\omega_1-\omega_2-\omega_5-\omega_6+i\epsilon},$$ (48b)

$$\phantom{x}_{\,2}^{4\,5} \!\!\!\gtrless\!\!\!\begin{smallmatrix}5\\6\end{smallmatrix}\!\!\!\!\!-\!1 = \frac{1}{4}\left(\frac{\lambda}{2\pi R}\right)^2 \sum_{1,\ldots,6} \delta_{1,256}\delta_{34,56}\langle f|\phi_4^{(-)}\phi_3^{(-)}\phi_2^{(-)}\phi_1^{(+)}|i\rangle$$
$$\times \frac{1}{2\omega_5}\frac{1}{2\omega_6}\frac{1}{\omega_3+\omega_4-\omega_5-\omega_6+i\epsilon}.$$ (48c)

Here the indices $5,6$ label the internal lines, and we use the shorthand $\delta_{12,34} = \delta_{k_1+k_2,k_3+k_4}$. Note the difference between the energy denominators in Eqs. (48a) and (48b), and the difference in the symmetry factor in Eq. (48c) compared to the previous two diagrams. We also

---

[5]This is the product of the coefficient of the normal-ordered operator in Wick's theorem and the coefficient of the operator in Eq. (44) in the normal ordered operator. For example,

$$:\phi^4: = \left(\phi^{(+)}\right)^4 + 4\phi^{(-)}\left(\phi^{(+)}\right)^3 + 6\left(\phi^{(-)}\right)^2\left(\phi^{(+)}\right)^2 + 4\left(\phi^{(-)}\right)^3\phi^{(+)} + \left(\phi^{(-)}\right)^4.$$ (47)

have disconnected diagrams such as

$$\begin{aligned}
&= \frac{1}{32}\left(\frac{\lambda}{2\pi R}\right)^3 \sum_{1,\dots,10} \delta_{12,910}\delta_{910,56}\delta_{34,78}\langle f|\phi_8^{(-)}\phi_7^{(-)}\phi_6^{(-)}\phi_5^{(-)}\phi_4^{(+)}\phi_3^{(+)}\phi_2^{(+)}\phi_1^{(+)}|i\rangle \\
&\qquad \times \frac{1}{2\omega_9}\frac{1}{2\omega_{10}}\frac{1}{\omega_5+\omega_6-\omega_9-\omega_{10}+i\epsilon} \\
&\qquad\qquad \times \frac{1}{\omega_5+\omega_6+\omega_7+\omega_8-\omega_3-\omega_4-\omega_9-\omega_{10}+i\epsilon}\,.
\end{aligned}$$
(48d)

The diagrammatic rules given above are for the fundamental theory. For the effective theory, the only difference is that the sums over intermediate states are restricted to the low-energy subspace. The intermediate states are associated with cuts between the vertices of the diagram. For each cut, we must include a step function that enforces the constraint that the total energy is below $E_{\max}$. Note that unlike the energy differences in the denominators, this constraint depends on the particles in the initial and final states that do not interact. For example, the step functions for the diagrams in Eqs. (48) evaluated for the effective theory are given by

$$: \Theta(E_{\max}-E_f+\omega_3+\omega_4-\omega_5-\omega_6)\,,$$
(49a)

$$: \Theta(E_{\max}-E_f-\omega_1-\omega_2-\omega_5-\omega_6)\,,$$
(49b)

$$: \Theta(E_{\max}-E_f+\omega_3+\omega_4-\omega_5-\omega_6)\,,$$
(49c)

$$\begin{aligned}
&: \Theta(E_{\max}-E_f+\omega_5+\omega_6-\omega_9-\omega_{10}) \\
&\qquad \times \Theta(E_{\max}-E_f+\omega_5+\omega_6+\omega_7+\omega_8-\omega_3-\omega_4-\omega_9-\omega_{10})\,.
\end{aligned}$$
(49d)

Recall that $E_f$ is the total energy of the final state, including the energy of particles that do not contract with any vertex (which are not drawn in the diagrams). The fact that diagrams depend on the energies of particles that do not participate in the interaction is a manifestation of the non-locality of the effective Hamiltonian.

# 4 Renormalization of 2D $\lambda\phi^4$ Theory

In this section, we discuss the renormalization of 2D $\lambda\phi^4$ theory. In this theory, the coupling $\lambda$ has dimensions of mass-squared, see Eq. (29). The theory is therefore super-renormalizable, and in fact all UV divergences can be eliminated by normal-ordering. We will however also consider more general regulators and renormalization schemes. We do this for several reasons. First, we will see below that separation of scales is manifest only in a more general renormalization scheme for the fundamental theory. Second, we wish to emphasize that renormalization of the fundamental theory can be carried out independently of the Hamiltonian truncation. This point will be important for theories with genuine UV divergences, such as the 3D $\lambda\phi^4$ theory considered in Appendix A.

## 4.1 Regularization

Matching the results of the renormalized fundamental theory onto the effective Hamiltonian requires considering the theory in finite volume. We compactify the spatial direction on a circle, which breaks Lorentz invariance and implies that particles carry discrete spatial momenta. We regularize the theory using a hard momentum cutoff on the spatial momenta:

$$\sum_k \quad \to \quad \sum_{k \le \Lambda R}, \tag{50}$$

where $\Lambda \gg E_{\max}$; we will eventually take $\Lambda \to \infty$. We choose this cutoff because makes it straightforward to compute the quantity $T$ that we use to match to the effective theory. The fact that this cutoff breaks Lorentz invariance does not cause any significant complication for the simple 2D $\lambda\phi^4$ model studied here, as we will see below.[6]

## 4.2 UV Divergences

Unlike the energy cutoff $E_{\max}$ that we impose on the effective Hamiltonian, the cutoff $\Lambda$ on the fundamental theory is a local Wilsonian cutoff. Therefore, the possible UV divergences can be classified by writing the possible local counterterms, using dimensional analysis to determine the dependence on the cutoff $\Lambda$. In 2D $\lambda\phi^4$ theory, the field $\phi$ is dimensionless. However, at $O(\lambda^n)$ in perturbation theory, loops can generate counterterms with at most $2n$ external $\phi$ lines. Because the cutoff breaks Lorentz invariance, we have to allow for the possibility of Lorentz violating counterterms. The leading counterterms have the schematic form

$$\Delta\mathcal{L} \sim \int \mathrm{d}^2x \left( \lambda \ln\Lambda + \lambda \ln\Lambda \phi^2 + \frac{\lambda^2}{\Lambda^2}(\phi^2 + \phi^4) + \frac{\lambda^2}{\Lambda^4}\left[(\partial_t\phi)^2 + \cdots\right] + \cdots \right). \tag{51}$$

We see that only the vacuum energy and $\phi^2$ mass term receive UV divergent contributions. The divergent vacuum energy in this theory means that only energy differences are physically meaningful.

We see that renormalizing the theory only requires introducing a bare mass parameter. Therefore, we write the bare Lagrangian as

$$\mathcal{L} = \frac{1}{2}(\partial\phi)^2 - \frac{1}{2}m_0^2\phi^2 - \frac{\lambda}{4!}\phi^4. \tag{52}$$

## 4.3 Renormalization

The fundamental and the effective theory must be defined on the same Fock space in order to carry out the matching described in §2.3. The unperturbed Hamiltonian for both theories is therefore given by Eq. (33), where $m_Q$ is a finite quantization mass that can be treated as a variational parameter to improve convergence [8]. The interaction term is then given by Eq. (35) with the coefficient of $:\phi^2:$ given by

$$m_V^2 = m_0^2 - m_Q^2 + \frac{\lambda}{8\pi R} \sum_{|k| \le \Lambda R} \frac{1}{\omega_k}. \tag{53}$$

---

[6]For more complicated models, it may be worthwhile to develop the technology for using Lorentz invariant cutoffs (such as dimensional regularization) to compute $T$ in finite volume.

Here $m_V^2$ is a finite, renormalized quantity.[7] This definition of the renormalized mass is convenient for calculations, since $m_V^2$ is the mass vertex that appears in our diagrammatic rules (see Eqs. (35) and (42)). However, we will see that separation of scales is manifest only in a different renormalization scheme where the couplings are renormalized at a renormalization scale $\mu \sim E_{\max}$.

It is therefore also useful to define a renormalized mass $m_R^2$ that depends on an arbitrary renormalization scale $\mu$:

$$m_R^2(\mu) = m_0^2 + \frac{\lambda}{8\pi R} \sum_{\mu R < |k| \leq \Lambda R} \frac{1}{\omega_k}. \tag{55}$$

This is the standard definition of the renormalized mass parameter, in which the contribution of modes with $k > \mu R$ have been absorbed into the renormalized coupling. We will see below that separation of scales is manifest in terms of $m_R^2(\mu \sim E_{\max})$. The mass parameter $m_V^2$ is related to $m_R^2$ by

$$m_V^2 = m_R^2(\mu) - m_Q^2 + \frac{\lambda}{8\pi R} \sum_{|k| \leq \mu R} \frac{1}{\omega_k}. \tag{56}$$

Previous work on Hamiltonian truncation of this model works directly with the normal-ordered Hamiltonian in which case there are no UV divergences, and hence no renormalization is needed. In our approach this corresponds to the choice $m_V^2 = 0$, so that $V$ consists only of a normal-ordered $:\phi^4:$ term. We then define the 'normal ordered mass' by

$$m_{NO} = m_R(\mu = 0). \tag{57}$$

Using Eq. (53) and Eq. (55), we see that this implies $m_Q = m_{NO}$ since $m_V^2 = 0$. (To eliminate the $k = 0$ mode from the sum in Eq. (56), $\mu$ must actually be taken to be slightly negative.) The normal ordered mass $m_{NO}$ is a convenient renormalization group invariant parameter we can use to define the theory.

# 5 Matching in 2D $\lambda\phi^4$ Theory

Now that we have renormalized the UV theory, we can turn to applying the general formalism given in Eq. (25) to derive the effective theory matching corrections. In this section, we will compute the leading terms in the low-energy expansion of the effective Hamiltonian at $O(V^2)$ in 2D $\lambda\phi^4$ theory. In this approximation, the effective Hamiltonian is local. In §6 we will explain the power counting in powers of $1/E_{\max}$ for this theory, and we will argue that these calculations give the leading $1/E_{\max}^2$ corrections to the effective Hamiltonian. The resulting effective Hamiltonian is therefore expected to have errors of order $1/E_{\max}^3$, which is confirmed by our numerical results in §7.

The calculations are a straightforward application of the diagrammatic rules presented in §3 above, but they illustrate some non-trivial features of the effective theory matching. In particular, matching at $O(V^2)$ involves 2- and 3-loop diagrams with overlapping UV/IR sensitivity. We will show that the IR sensitivity of the corrections is canceled by other contributions

---

[7]That is, the log UV divergence in the sum

$$\sum_{|k| \leq \Lambda R} \frac{1}{\omega_k} \sim \int^\Lambda \frac{dk}{\omega_k} \sim \ln \Lambda \tag{54}$$

is canceled by allowing the bare mass $m_0^2$ to depend on the cutoff.

as required by separation of scales in the effective theory, but this cancellation requires both the correct renormalization prescription and definition of the operators in the effective Hamiltonian.

## 5.1 Matching at $O(V)$: Operator Approach

We begin at $O(V)$; the matching condition is simply given by (see Eq. (25a))

$$\langle f | H_1 | i \rangle_{\text{eff}} = \langle f | V | i \rangle. \tag{58}$$

We must be careful in interpreting this expression, since the left-hand side is evaluated in the effective theory with a truncated Hilbert space. The correct way to match is to equate the coefficients of normal ordered operators in the fundamental and effective theories. This approach agrees with Eq. (58), because at this order the states above $E_{\text{max}}$ only impact the counterterm and normal ordering constants. The normal-ordered full theory potential is given by Eq. (35), with $m_V^2$ (the coefficient of $:\phi^2:$) given by Eq. (53).

The $O(V)$ correction to the effective theory has the form[8]

$$H_1 = \int dx \left[ \frac{1}{2} m_1^2 \phi^2 + \frac{\lambda_1}{4!} \phi^4 \right]$$

$$= \int dx \left[ \frac{1}{2} m_{V1}^2 :\phi^2: + \frac{\lambda_1}{4!} :\phi^4: \right] + \text{constant}, \tag{60}$$

with

$$m_{V1}^2 = m_1^2 + \frac{\lambda_1}{8\pi R} \sum_{|k| \leq k_{\text{max}}} \frac{1}{\omega_k}, \tag{61}$$

where

$$k_{\text{max}} = R\sqrt{E_{\text{max}}^2 - m_Q^2} \simeq E_{\text{max}} R. \tag{62}$$

Note that the normal-ordering in Eq. (60) is performed on the truncated Hilbert space. We drop normal-ordering constants proportional to the identity operator, since they do not contribute to differences of energy eigenvalues. (We will, however, need these contributions when we discuss the separation of scales for the vacuum diagrams in §5.6.)

Matching the coefficients of $:\phi^4:$ and $:\phi^2:$ then gives simply

$$\lambda_1 = \lambda, \qquad m_{V1}^2 = m_V^2. \tag{63}$$

Note that $m_{V1}^2$ is finite, as all couplings in the effective theory must be, since we are matching onto a renormalized fundamental theory.

We now discuss the separation of scales at this order. Separation of scales should not apply to the coefficients of the normal ordered operators, since normal ordering depends on the split of the full Hamiltonian $H$ into 'free' and 'interacting' terms. Instead, we expect that separation of scales will hold for the coefficients of non-normal-ordered operators. Furthermore, effective field theory methodology tells us that separation of scales should be manifest in terms of

---

[8]Here we use the expressions

$$\phi^2 = :\phi^2: + Z, \qquad \phi^4 = :\phi^4: + 6Z:\phi^2: + 3Z^2, \tag{59}$$

with $Z = \frac{1}{4\pi R} \sum_k \frac{1}{\omega_k}$ for the 2D $\lambda\phi^4$ theory.

couplings renormalized at the matching scale, in this case $\mu \sim E_{\text{max}}$. To see that these expectations are satisfied at $O(V)$, note that the coefficient of $\phi^2$ in the effective theory at this order is given by

$$m_Q^2 + m_1^2 = m_R^2(\mu = k_{\text{max}}/R). \tag{64}$$

We will see a much more nontrivial check of separation of scales when we consider 2- and 3-loop diagrams in §5.5 and §5.6 below.

## 5.2 Matching at $O(V)$: Diagrammatic Approach

Although the operator approach to the matching at $O(V)$ is very simple, it becomes more cumbersome at higher orders. This motivates us to re-derive it in terms of diagrams for illustration, since we will rely on the diagrammatic approach to compute the $O(V^2)$ corrections below. Recall that we defined the diagrammatic expansion so that the vertices are given by the coefficients of the normal-ordered operators; contractions (as defined using Wick's theorem) between the same vertex are omitted, see §3. The $O(V)$ contributions to $\langle f|T|i\rangle$ in the fundamental theory include the 4-point tree diagrams

$$\times \; + \; {\scriptstyle >}\!\!-\!\! + \; -\!\!{\scriptstyle <} \; + \; {\scriptstyle >}\!\!\!\!> \; + \; {\scriptstyle <}\!\!\!\!< \; = \; \frac{\lambda}{4!} \int \mathrm{d}x \, \langle f|:\phi^4:|i\rangle. \tag{65}$$

For contact interaction diagrams such as these, the different ways of contracting the external lines to initial and final states add together to build up the full contribution of the local operator. In addition, there are diagrams given by the 2-point vertex (see Eq. (42)). Recalling that the 2-point vertex is given by the coefficient of $:\phi^2:$, we have for the fundamental theory

$$-\!\!\!\bullet\!\!\!- \; + \; {\scriptstyle >}\!\!\!\bullet \; + \; \bullet\!\!{\scriptstyle <} \; = \; \frac{1}{2}m_V^2 \int \mathrm{d}x \, \langle f|:\phi^2:|i\rangle. \tag{66}$$

In the effective theory, we have the same diagrams, with $\lambda \to \lambda_1$, and $m_V^2 \to m_{V1}^2$. This reproduces Eq. (63), as it must.

## 5.3 Matching at $O(V^2)$: 4 Legs

We now consider the matching at $O(V^2)$ using the diagrammatic approach. We begin with diagrams that have 4 external legs, because they are simpler to evaluate than the 2-point corrections. The diagrams that contribute are given by

$$\langle f|T_{2,4}|i\rangle = \;{>}\!\!\!\infty\!\!\!{<} \; + \; \bigotimes \; + \; {>}\!\!\!\circ\!\!\!\diagup$$

$$+ \; {>}\!\!\!\circ\!\!-\!\! \; + \; -\!\!{\circ}\!\!\!{<} \; + \; {>}\!\!\!\circ\!\!\!> \; + \; {\circ}\!\!\!{<} \;. \tag{67}$$

Note that there are no diagrams with $\phi^2$ vertices at this order.

To match, we equate the diagrams in Eq. (67) in the fundamental and effective theories. This means that the effective Hamiltonian depends on the difference between the diagrams computed using the two descriptions of the theory. For the first diagram on the right-hand side of Eq. (67), the difference is given by (see Eq. (48a))

$$\substack{3 \\ 4}\!\!\overset{5}{\underset{6}{>\!\!\!\infty\!\!\!<}}\!\!\substack{1 \\ 2} - \left[\substack{3 \\ 4}\!\!\overset{5}{\underset{6}{>\!\!\!\infty\!\!\!<}}\!\!\substack{1 \\ 2}\right]_{\text{eff}} = \frac{\lambda^2}{128\pi^2 R^2} \sum_{1,\dots,4} \delta_{12,34} \langle f|\phi_4^{(-)}\phi_3^{(-)}\phi_2^{(+)}\phi_1^{(+)}|i\rangle$$

$$\times \sum_{5,6} \delta_{56,34} \frac{\Theta(\omega_5 + \omega_6 - \omega_3 - \omega_4 + E_f - E_{\text{max}})}{\omega_5 \omega_6(\omega_3 + \omega_4 - \omega_5 - \omega_6)}. \tag{68}$$

We are omitting the $i\epsilon$ factors, which we showed do not affect the matching in §2. Also, note that we are no longer explicitly including the UV cutoff $\Lambda$ in the sums over momenta. Effective field theory methodology tells us that the effective Hamiltonian should be determined by matching low-energy observables and expanding in powers of IR scales [25]. In our case, the IR scales are given by

$$m_Q \lesssim \omega_{1,2,3,4} \lesssim E_{i,f} \ll E_{\max}. \tag{69}$$

The step function in Eq. (68) then implies that

$$\omega_{5,6} \gtrsim E_{\max}. \tag{70}$$

That is, the matching is only sensitive to intermediate states whose energies are above the cutoff of the truncated theory. This is a manifestation of the separation of scales in effective field theory. The full theory diagram has a complicated dependence on the external energies and momenta, but because we are matching matrix elements of low-lying states ($E_{i,f} \ll E_{\max}$), this dependence can be expanded in a power series. The first term that results from taking this expansion can be determined by setting all of the external momenta and energies to zero. In this approximation, we have

$$\left[{}^3_4 \!\!\!\bowtie^5_6 {}^1_2\right] - \left[{}^3_4 \!\!\!\bowtie^5_6 {}^1_2\right]_{\rm eff} \simeq -\frac{\lambda^2}{128\pi R} \int dx \, \langle f | [\phi^{(-)}]^2 [\phi^{(+)}]^2 | i \rangle$$
$$\times \sum_k \frac{\Theta(2\omega_k - E_{\max})}{\omega_k^3} \Big[ 1 + O(E_{i,f}/E_{\max}) \Big], \tag{71}$$

where we used

$$\sum_{1,\dots,4} \delta_{12,34} \langle f | \phi_4^{(-)} \phi_3^{(-)} \phi_2^{(+)} \phi_1^{(+)} | i \rangle = 2\pi R \int dx \, \langle f | [\phi^{(-)}]^2 [\phi^{(+)}]^2 | i \rangle . \tag{72}$$

We see that in this approximation, the matrix element of the correction is a local operator. We therefore refer to this as the 'local approximation.'

The other diagrams in Eq. (67) differ from the one considered in Eq. (71) by its initial and final state contractions, and its energy denominators, and (for the effective theory) the dependence of the $\Theta$ functions that implement the cutoff. But in the leading approximation discussed above, the latter two effects disappear. The diagrams therefore differ only by the matrix elements of fields associated with the external lines. Summing over all diagrams with all possible choices for the external lines builds up the normal-ordered operator, as we found for the diagrammatic matching at $O(V)$ in §5.2. Putting it all together, we find

$$\langle f | T_{2,4} | i \rangle - \langle f | T_{2,4} | i \rangle_{\rm eff} \simeq -\frac{\lambda^2}{128\pi R} \sum_k \frac{\Theta(2\omega_k - E_{\max})}{\omega_k^3} \int dx \, \langle f | {:}\phi^4{:} | i \rangle . \tag{73}$$

This gives a contribution to the effective Hamiltonian

$$H_{2,4} \simeq \frac{\lambda_2}{4!} \int dx \, {:}\phi^4{:} \,, \tag{74}$$

where

$$\lambda_2 = -\frac{3\lambda^2}{16\pi R} \sum_k \frac{\Theta(2\omega_k - E_{\max})}{\omega_k^3} . \tag{75}$$

Note that $\lambda_2$ depends only on the properties of states above the cutoff, and therefore obeys the principle of separation of scales. The sum is UV convergent, so it is dominated by the contribution of states near the cutoff.

For illustration, we compute the sum in Eq. (75) in the large $E_{\max}$ limit by approximating it as an integral:

$$\lambda_2 \simeq -\frac{3\lambda^2}{16\pi R} \times 2 \int_{\frac{1}{2}E_{\max}R}^{\infty} \frac{dk}{\omega_k^3} = -\frac{3\lambda^2}{4\pi E_{\max}^2}\Big[1 + O\big(m^2/E_{\max}^2\big)\Big]. \qquad (76)$$

Note that this has the form we expect from general power counting arguments (see Eq. (101)). We can systematically correct the approximation Eq. (76) using the Euler-MacLaurin summation formula Eq. (98) to include higher orders in the IR scales $R^{-1}/E_{\max}$ and $m_Q/E_{\max}$. To obtain numerical results in §7, we will directly evaluate sums such as Eq. (75) numerically.

### 5.4 Matching at $O(V^2)$: 2 Legs

Now we consider the matching contribution at $O(V^2)$ from diagrams with 2 external legs:

$$\langle f | T_{2,2} | i \rangle = \;\;\text{[diagrams]}\;\; . \qquad (77)$$

Let us begin with the first 2-loop diagram shown in Eq. (77). As we discussed in the previous section, matching is determined by the difference between the diagrams in the fundamental and the effective theory:

$$\text{[diagram]} - \Big[\text{[diagram]}\Big]_{\text{eff}} = \frac{1}{6}\left(\frac{\lambda}{2\pi R}\right)^2 \sum_{1,\dots,5} \delta_{1,2}\delta_{1,345}\langle f | \phi_2^{(-)}\phi_1^{(+)} | i \rangle$$
$$\times \frac{1}{2\omega_3}\frac{1}{2\omega_4}\frac{1}{2\omega_5}\frac{\Theta(\omega_3 + \omega_4 + \omega_5 - E_{\max})}{\omega_2 - \omega_3 - \omega_4 - \omega_5}. \qquad (78)$$

Note that, unlike the 1-loop diagrams analyzed in the previous subsection, the sum over the intermediate momenta $k_{3,4,5}$ includes mixed UV/IR regions where some of the momenta are large, while others are small. Therefore, the separation of UV and IR scales is not manifest diagram by diagram at this order.[9] However, we will show in §5.5 that this important property manifests when we add all the diagrams together and use the correct renormalization prescription and definition of operators. This justifies our use of the approximation neglecting the external energies and momenta since they are much smaller than the intermediate momenta we are integrating out. As above, we refer to this simplifying choice as the 'local approximation.'

For the diagram above, the local approximation gives

$$\text{[diagram]} - \Big[\text{[diagram]}\Big]_{\text{eff}} \simeq \frac{\lambda^2}{192\pi^2 R^2} \int dx \, \langle f | \phi_2^{(-)}\phi_1^{(+)} | i \rangle$$
$$\times \sum_{3,4,5} \delta_{345,0} \frac{\Theta(\omega_3 + \omega_4 + \omega_5 - E_{\max})}{\omega_3 \omega_4 \omega_5 (-\omega_3 - \omega_4 - \omega_5)}. \qquad (79)$$

---

[9]In the context of renormalization theory, this is the problem of overlapping divergences.

Note that the dependence on the external momenta factors out, implying that the matrix element is local. The first 4 diagrams in Eq. (77) are the same up to the factors that depend on the initial and final states, and combining them gives

$$
\bigodot + \bigodot + \bigodot + \overline{\bigodot} - \Big[ \bigodot + \bigodot + \bigodot + \overline{\bigodot} \Big]_{\text{eff}}
$$

$$
\simeq \frac{\lambda^2}{192\pi^2 R^2} \int dx \, \langle f| : \phi^2 : |i\rangle \times \sum_{3,4,5} \delta_{345,0} \frac{\Theta(\omega_3 + \omega_4 + \omega_5 - E_{\text{max}})}{\omega_3 \omega_4 \omega_5 (-\omega_3 - \omega_4 - \omega_5)} \, . \tag{80}
$$

Finally, we compute the remaining 6 diagrams in Eq. (77) in the local approximation:

$$
\bigcirc\!\!\!\!\!\diagup + \!\!\diagup\!\!\!\!\!\bigcirc\!\!\!\!\!\diagup + \cdots - \Big[ \bigcirc\!\!\!\!\!\diagup + \!\!\diagup\!\!\!\!\!\bigcirc\!\!\!\!\!\diagup + \cdots \Big]_{\text{eff}}
$$

$$
\simeq -\frac{\lambda}{32\pi R} \int dx \, \langle f| : \phi^2 : |i\rangle \Big[ m_V^2 \sum_k \frac{1}{\omega_k^3} - m_{V1}^2 \sum_{|k| \le k_{\text{max}}} \frac{\Theta(E_{\text{max}} - 2\omega_k)}{\omega_k^3} \Big]
$$

$$
= -\frac{\lambda m_V^2}{32\pi R} \int dx \, \langle f| : \phi^2 : |i\rangle \sum_k \frac{\Theta(2\omega_k - E_{\text{max}})}{\omega_k^3} \, . \tag{81}
$$

As above, combining the diagrams together yields the matrix element of the $:\phi^2:$ operator. We have used the $O(V)$ matching condition given in Eq. (63) to derive the last line. Combining this with Eq. (80), we obtain (in the local approximation)

$$
H_{2,2} \simeq \frac{1}{2} m_{V2}^2 \int dx : \phi^2 : \, , \tag{82}
$$

where

$$
m_{V2}^2 = \frac{\lambda}{16\pi R} \Big[ \frac{\lambda}{6\pi R} \sum_{3,4,5} \delta_{345,0} \frac{\Theta(\omega_3 + \omega_4 + \omega_5 - E_{\text{max}})}{\omega_3 \omega_4 \omega_5 (-\omega_3 - \omega_4 - \omega_5)} - m_V^2 \sum_k \frac{\Theta(2\omega_k - E_{\text{max}})}{\omega_k^3} \Big] . \tag{83}
$$

These expressions are quite complicated, so for numerical studies we numerically evaluate the sums in the matching corrections, making sure that the sums have adequately converged. The matching correction is dominated by the contribution of states near the $E_{\text{max}}$ cutoff, so we believe it is important to treat these states accurately in the matching, as we have done here.

## 5.5 Separation of Scales at $O(V^2)$: 2 Legs

The result Eq. (83) does not appear to satisfy the separation of scales principle in the effective theory, since both terms contain sums with arbitrarily small momenta (since $m_V^2$ contains an IR sum, see Eq. (56)). The expectation based on effective field theory methodology is that the cancelation of such overlapping UV/IR regions takes place only if all diagrams are included, and if the renormalization scale is chosen to be near the cutoff of the effective theory [25]. As already discussed in §5.1, the normal ordered operators have a renormalization scale near the IR, and so separation of scales applies to the coefficients of non-normal ordered operators defined with a renormalization scale $\mu \sim E_{\text{max}}$. We therefore write

$$
H_2 = \int dx \Big[ \frac{1}{2} m_2^2 \phi^2 + \frac{\lambda_2}{4!} \phi^4 \Big] , \tag{84}
$$

where (see Eqs. (74), (82) and (59))

$$m_2^2 = m_{V2}^2 - \frac{\lambda_2}{8\pi R} \sum_{|k| \le k_{\text{max}}} \frac{1}{\omega_k}. \tag{85}$$

Our goal is to show that all the contributions to $m_2^2$ that involve products of UV and IR dominated sums cancel when all the contributions are included. For example, the sum in the first line of Eq. (83) includes a region where one of $k_{1,2,3}$ is much smaller than the other two:

$$\sum_{3,4,5} \delta_{345,0} \frac{\Theta(\omega_3 + \omega_4 + \omega_5 - E_{\text{max}})}{\omega_3 \omega_4 \omega_5 (-\omega_3 - \omega_4 - \omega_5 + i\epsilon)} \simeq 3 \sum_{|k'| \ll k_{\text{max}}} \frac{1}{\omega_{k'}} \times \sum_k \frac{\Theta(2\omega_k - E_{\text{max}})}{\omega_k^2 (-2\omega_k)} + \cdots. \tag{86}$$

The factor of 3 comes from the 3 distinct regions where one of the $k_{1,2,3}$ is small. We see that all of the mixed UV/IR contributions to $m_2^2$ have the same factorized form:

$$m_2^2 \simeq -\frac{\lambda^2}{32\pi^2 R^2} \underbrace{\left( \frac{1}{2} + \frac{1}{4} - \frac{3}{4} \right)}_{=0} \sum_{|k'| \ll k_{\text{max}}} \frac{1}{\omega_{k'}} \times \sum_k \frac{\Theta(2\omega_k - E_{\text{max}})}{\omega_k^3} + \cdots. \tag{87}$$

Here, the first two terms in the parentheses are contributions to $m_2^2$, and the third comes from un-normal-ordering $\phi^4$ (see Eq. (59)). The fact that the mixed UV/IR terms cancel shows that the 2-loop correction to the 2-point function does in fact manifest the separation of scales. Not only is this a critical check that our matching procedure is well defined in the limit that the IR scales are taken to zero, it also justifies our use of the local approximation in the matching calculation above.[10]

Note that the contribution from states below the cutoff does not cancel exactly in the sums above. Separation of scales requires only that the dominant contribution to effective couplings comes from states *near* the cutoff, and this is what we have demonstrated above.

## 5.6 Matching and Separation of Scales at $O(V^2)$: 0 Legs

In this section, we consider the vacuum diagrams at $O(V^2)$. Recall that our diagrammatic rules are defined so that there are no contractions between lines at the same vertex, so there are only 2 diagrams:

$$\langle f | T_{2,0} | i \rangle = \vcenter{\hbox{⬭}} + \vcenter{\hbox{◉}}. \tag{88}$$

In the local approximation, these yield a contribution proportional to the identity operator in the effective Hamiltonian. We will compute these diagrams to give an additional test of the separation of scales.

The 3-loop vacuum diagram in the local approximation gives a matching contribution

$$\vcenter{\hbox{⬭}} - \left[ \vcenter{\hbox{⬭}} \right]_{\text{eff}} = \langle f | i \rangle \frac{1}{24} \left( \frac{\lambda}{2\pi R} \right)^2 \sum_{1,\dots,4} \delta_{1234,0} \frac{\Theta(\omega_1 + \omega_2 + \omega_3 + \omega_4 - E_{\text{max}})}{16\omega_1 \omega_2 \omega_3 \omega_4 (-\omega_1 - \omega_2 - \omega_3 - \omega_4)}. \tag{89}$$

This diagram contains two different kinds of regions with overlapping UV/IR dominated sums: one where 2 momenta are large and 2 are small (with multiplicity 6) and one where 3 momenta

---

[10]It would be very interesting to check the separation of scales beyond the local approximation.

are large and 1 is small (with multiplicity 4). These are given by

$$\bigcirc\!\!\!\bigcirc\!\!\!\bigcirc - \left[\bigcirc\!\!\!\bigcirc\!\!\!\bigcirc\right]_{\text{eff}} \simeq \langle f|i\rangle \frac{6}{24}\left(\frac{\lambda}{2\pi R}\right)^2 \sum_k \frac{\Theta(2\omega_k - E_{\max})}{16\omega_k^2(-2\omega_k)} \sum_{|k'|,|k''|\ll k_{\max}} \frac{1}{\omega_{k'}\omega_{k''}}$$

$$+ \langle f|i\rangle \frac{4}{24}\left(\frac{\lambda}{2\pi R}\right)^2 \sum_k \frac{\Theta(2\omega_k + \omega_{2k} - E_{\max})}{16\omega_k^2\omega_{2k}(-2\omega_k - \omega_{2k})} \sum_{|k'|\ll k_{\max}} \frac{1}{\omega_{k'}}$$

$$+ \cdots . \tag{90}$$

The first term is $\sim (\ln m_Q)^2/E_{\max}^2$, while the second is $\sim (\ln m_Q)/E_{\max}^3$.

The 1-loop vacuum diagram gives a matching contribution

$$\bigcirc\!\!\!\!\!\bullet - \left[\bigcirc\!\!\!\!\!\bullet\right]_{\text{eff}} = \langle f|i\rangle \frac{1}{2}\left(m_V^2\right)^2 \sum_k \frac{\Theta(2\omega_k - E_{\max})}{4\omega_k^2(-2\omega_k)} . \tag{91}$$

We expect separation of scales to be manifest only if we use a renormalization scheme in the fundamental theory where $\mu \sim E_{\max}$. In this case, Eq. (91) has an overlapping UV/IR dominated region because $m_V^2$ is IR dominated (see Eq. (56)). The overlapping region gives a contribution $\sim (\ln m_Q)^2/E_{\max}^2$.

As we have previously discussed, we also expect separation of scales to be manifest for the effective Hamiltonian written in terms of operators that are not normal ordered. This effective Hamiltonian has additional contributions to the identity operator from un-normal-ordering the matching contributions computed above using Eq. (59). Writing

$$H_{2,0} = C_2 \int dx \, \mathbb{1} , \tag{92}$$

the additional contributions to $C_2$ from un-normal-ordering are

$$\Delta C_2 = \left(\frac{\lambda}{2\pi R}\right)^2 \frac{1}{16}\frac{1}{4\pi R}\left[\frac{1}{3}\sum_k \frac{\Theta(2\omega_k + \omega_{2k} - E_{\max})}{\omega_k^2\omega_{2k}(2\omega_k + \omega_{2k})}\sum_{|k'|\ll k_{\max}} \frac{1}{\omega_{k'}}\right.$$

$$\left. + \frac{3}{8}\sum_k \frac{\Theta(2\omega_k - E_{\max})}{\omega_k^3}\sum_{|k'|,|k''|\ll k_{\max}} \frac{1}{\omega_{k'}\omega_{k''}}\right] + \cdots . \tag{93}$$

Combining these contributions, the overlapping UV/IR contributions to $C_2$ are given by

$$C_2 \simeq \frac{\lambda^2}{128\pi^2 R^2}\left(\underbrace{\frac{1}{3} - \frac{1}{3}}_{=0}\right)\sum_k \frac{\Theta(2\omega_k + \omega_{2k} - E_{\max})}{\omega_k^2\omega_{2k}(2\omega_k + \omega_{2k})}\sum_{|k''|\ll k_{\max}} \frac{1}{\omega_{k''}}$$

$$+ \frac{\lambda^2}{128\pi^2 R^2}\left(\underbrace{\frac{3}{8} - \frac{1}{8} - \frac{1}{4}}_{=0}\right)\sum_k \frac{\Theta(2\omega_k - E_{\max})}{\omega_k^3}\sum_{|k|,|k''|\ll k_{\max}} \frac{1}{\omega_{k'}\omega_{k''}} + \cdots . \tag{94}$$

The positive contributions are from Eq. (93), and the negative contributions are from the diagrams in Eqs. (90) and (91). Once again, we see that overlapping UV/IR regions cancel, leaving a effective coefficient that is dominated by the contributions of states near the cutoff.

# 6 Power Counting and Locality

The idea of Hamiltonian truncation is that the low-lying states of the system can be well-approximated by a truncated Hilbert space with maximum energy (measured by $H_0$) $E_{\max}$.

Effective field theory ideas suggest that we should therefore be able to compute low-energy observables in a systematic expansion in powers of $1/E_\text{max}$. In this section, we present a power-counting scheme that we believe gives the general form of the effective Hamiltonian defined above. An important feature of this power counting is that the non-locality of the effective Hamiltonian is controlled in the $1/E_\text{max}$ expansion.

Our discussion will be for 2D $\lambda\phi^4$ theory, but it is straightforward to generalize to other theories. When we present numerical results in §7, we will see that the errors are consistent with the predictions of the power counting presented here.

## 6.1 Non-locality and Non-Hermiticity at $O(V^2)$

Before presenting the general power counting, we present calculations of non-local (and non-Hermitian) contributions to the effective Hamiltonian. This will help motivate the general power counting we present in the following subsection.

We will compute the $O(E_f)$ correction to the matching contribution in Eq. (68). To simplify the calculation, we assume that

$$\omega_{1,2,3,4} \ll E_{i,f} \ll E_\text{max}, \tag{95}$$

so the sum over internal momenta in Eq. (68) can be approximated by

$$\sum_{5,6} \delta_{56,34} \frac{\Theta(\omega_5 + \omega_6 - \omega_3 - \omega_4 + E_f - E_\text{max})}{\omega_5 \omega_6 (\omega_3 + \omega_4 - \omega_5 - \omega_6)} \simeq \sum_k \frac{\Theta(2\omega_k + E_f - E_\text{max})}{\omega_k^2(-2\omega_k)}. \tag{96}$$

The approximation Eq. (95) allows us to neglect the masses, so the sum over $k$ can be taken over the range

$$|k| > \frac{(E_\text{max} - E_f)R}{2}. \tag{97}$$

This can be evaluated as a series in $1/E_\text{max}$ using the Euler-Maclauren summation formula

$$\sum_{k=k_\text{min}}^{\infty} f(k) = \int_{k_\text{min}}^{\infty} dk\, f(k) + \tfrac{1}{2} f(k_\text{min}) - \sum_{r=1}^{\infty} \frac{B_{2r}}{(2r)!} f^{(2r)}(k_\text{min}), \tag{98}$$

where $B_{2r}$ are the Bernoulli numbers. Eq. (98) is an asymptotic expansion valid for any sufficiently smooth function $f$ such that $f(k)$ and its derivatives vanish as $k \to \infty$. Applying this to the sum in Eq. (96) gives

$$\sum_k \frac{\Theta(2\omega_k + E_f - E_\text{max})}{\omega_k^3} = \frac{4R}{E_\text{max}^2}\left[1 + \frac{2E_f}{E_\text{max}} + O(1/E_\text{max}R) + O(E_f/E_\text{max}^2 R) + O(E_f^2/E_\text{max}^2)\right]. \tag{99}$$

There are similar contributions from the other diagrams. Including the $O(E_f)$ corrections, the effective Hamiltonian has a contribution (compare to Eq. (74))

$$H_{2,4} = \frac{\lambda_2}{4!} \int dx \left[ :\phi^4: + \frac{2H_0}{E_\text{max}} :\phi^4: + \cdots \right], \tag{100}$$

where $\lambda_2$ is given by Eq. (75). Note that the $E_f$ dependence has been written in terms of $H_0$. The term in $H_{2,4}$ that depends on $H_0$ is non-local, because $H_0$ is itself given by an integral over $x$. From the diagrammatic expansion, we expect that all of the non-locality arises from the energy dependence in the step functions that define the cutoff of the effective Hamiltonian. Expanding these in powers of the energies of the external states gives powers of $H_0$, as in this example. Therefore, we expect that *all* of the non-locality of the effective Hamiltonian can be parameterized by powers of $H_0/E_\text{max}$. Note that the non-local term in Eq. (100) is also not Hermitian, which is consistent with the fact that $H_\text{eff}$ is non-Hermitian at $O(V^2)$ (see Eq. (25b)).

## 6.2 General Form of the Effective Hamiltonian

We now discuss what we expect for the general form of the effective Hamiltonian. The cutoff on the effective theory is defined by $H_0$, and this is the only source of non-locality in the theory. We therefore conjecture that all of the non-locality in the effective theory can be parameterized by the free Hamiltonian $H_0$. This leads naturally to the following general form for the $O(V^n)$ term in the effective Hamiltonian:

$$H_n \sim \frac{\lambda^n}{E_{\max}^{2n-2}} \sum_a \int \mathrm{d}x \, C_{na}\left( \frac{H_0}{E_{\max}}, \frac{R^{-1}}{E_{\max}}, \frac{m_{\mathrm{Q}}}{E_{\max}} \right) \mathcal{O}_{na}\left( \frac{\partial_x}{E_{\max}}, \frac{\partial_t}{E_{\max}}, \phi \right), \tag{101}$$

where we have factored out powers of $E_{\max}$ so that $C_{na}$ and $\mathcal{O}_{na}$ are dimensionless. Here $\mathcal{O}_{na}$ is a local Hermitian operator made of the field $\phi$ and its derivatives, and the real coefficient $C_{na}$ depends on the free Hamiltonian $H_0$, as well as the IR mass scales $m_{\mathrm{Q}}$ and $R^{-1}$. Expanding $\mathcal{O}_n$ in derivatives, and $C_n$ in powers of $m_{\mathrm{Q}}, R^{-1}$, and $H_0$ gives the effective Hamiltonian as an expansion in $1/E_{\max}$. Note that the powers of $H_0$ in Eq. (101) are to the left of the operator $\mathcal{O}_n$. There is no loss of generality in writing it this way, since $[H_0, \phi] = -\mathrm{i}\dot{\phi}$.

In fact, the form of the effective Hamiltonian is a bit more constrained than Eq. (101). The $\phi \mapsto -\phi$ symmetry implies that only even powers of $\phi$ can appear in $\mathcal{O}_n$. Second, parity invariance implies that only even powers of $\partial_x$ can appear in $\mathcal{O}_n$. Time reversal allows linear terms in time derivatives, since time reversal includes complex conjugation, so that $\mathrm{i}\partial_t$ is invariant. (Factors of $\mathrm{i}\partial_t$ in the effective Hamiltonian correspond to factors of single-particle energies $\omega_k$ in the transition amplitude.) The diagrammatic rules depend on $m_{\mathrm{Q}}$ only quadratically (through $\omega_k$), so $C_{na}$ involves only even powers of $m_{\mathrm{Q}}$. Finally, the number of powers of $\phi$ is limited by the fact that the interaction contains only 4 powers of the fields; for example, at $O(V^2)$ we have only $\phi^2$ and $\phi^4$ terms, as we have already seen above. Taking all this into account, the power counting rule predicts the parametric form

$$H_2 \sim \frac{\lambda^2}{E_{\max}^2} \int \mathrm{d}x \left[ \phi^2 + \phi^4 + \frac{R^{-1} + H_0}{E_{\max}}\left(1 + \phi^2 + \phi^4\right) + \frac{1}{E_{\max}}\left(\phi + \phi^3\right)\mathrm{i}\dot{\phi} + O(1/E_{\max}^2) \right]. \tag{102}$$

Eq. (101) gives $H_3 \sim \lambda^3/E_{\max}^4$, so the leading corrections to the effective theory beyond the order computed in §5.3 and §5.4 above comes from $1/E_{\max}^3$ corrections to $H_2$. These can be computed by going beyond the local approximation. This calculation is straightforward, but not trivial. For example, note that Eq. (102) contains a 3-loop contribution (from the diagram Eq. (89)) that is $\sim \lambda^2 H_0/E_{\max}^3$. Computing the full $1/E_{\max}^3$ corrections and checking that they further reduce the truncation error as indicated by the power counting is well worth doing. We plan to address this in future work.

# 7 Numerical Results for 2D $\lambda\phi^4$ Theory

In this section, we present numerical results for 2D $\lambda\phi^4$ theory using the improved effective Hamiltonian described above. Our main focus is on the convergence of the results as a function of the cutoff $E_{\max}$, rather than the determination of physically interesting quantities. We therefore choose the dimensionful parameters $\lambda$, $m^2$, and $R$ so that the theory has a single physical IR scale $M_{\mathrm{IR}}$. The only large parameter in the Hamiltonian truncation is then $E_{\max}/M_{\mathrm{IR}}$, where $E_{\max}$ is the cutoff on the Hamiltonian truncation. The dimensionful parameters $\lambda$, $m^2$, and $R$ are related to the physical IR scale $M_{\mathrm{IR}}$ by dimensional analysis. However, some of the dimensionless constants of proportionality are expected to differ significantly from 1, as we now explain.

## 7.1 Explanation of Dimensionless Factors

First, we consider the coupling $\lambda$, which has mass dimension 2. This means that it gets strong at a scale $M_{\text{IR}} \propto \sqrt{\lambda}$. To estimate the constant of proportionality, we use the idea that the scale $M_{\text{IR}}$ is the IR cutoff scale where perturbative loop corrections are the same size as tree-level effects ('naïve dimensional analysis') [28]. In 2D, the perturbative loop corrections are proportional to

$$\lambda \int \frac{\mathrm{d}^2 p}{(2\pi)^2} f(p^2) = \frac{\lambda}{4\pi} \int_0^\infty \mathrm{d}p^2 f(p^2). \tag{103}$$

(Note that $\lambda$ is defined to be the coefficient of $\phi^4/4!$ in the Lagrangian, so the symmetry factors in loop corrections are order 1.) Therefore we assume that the physical IR scale defined by the coupling $\lambda$ is given by

$$M_{\text{IR}}^2 \sim \frac{\lambda}{4\pi}. \tag{104}$$

Next, we consider the compactification radius $R$. In perturbation theory, the effect of this is to give a series of Kaluza-Klein excitations with energy differences $1/R$. We therefore choose

$$R^{-1} \sim M_{\text{IR}}. \tag{105}$$

Note that the finite-size corrections for particles of mass $m$ in 1 spatial dimension are proportional to $e^{-2\pi R m}$ [29], so this choice may be sufficient to neglect finite size effects.

Finally we consider the mass parameter $m^2$. At tree-level, it is clear that $M_{\text{IR}} \sim m$, but we must consider the possibility of large loop effects. However, as shown in §4, the renormalized coefficient of $\phi^2$ in the effective Hamiltonian is independent of the renormalization scheme used for the fundamental theory. To present our numerical results, we therefore mostly use the 'normal-ordered scheme' defined in §4.3, and we assume

$$m_{\text{NO}} \sim M_{\text{IR}}, \tag{106}$$

where $m_{\text{NO}}$ is the coefficient of $:\phi^2:$ in the fundamental theory.

## 7.2 Numerical Results

We now present the numerical results. Collecting the results above, the effective Hamiltonian is given by

$$H_{\text{eff}} = H_0 + \int \mathrm{d}x \left[ \frac{1}{2} (m_V^2 + m_{V2}^2) :\phi^2: + \frac{\lambda + \lambda_2}{4!} :\phi^4: \right], \tag{107}$$

where the corrections are calculated in §5.1 and §5.3:

$$m_{V2}^2 = \frac{\lambda}{16\pi R} \left[ \frac{\lambda}{6\pi R} \sum_{3,4,5} \delta_{345,0} \frac{\Theta(\omega_3 + \omega_4 + \omega_5 - E_{\max})}{\omega_3 \omega_4 \omega_5 (-\omega_3 - \omega_4 - \omega_5)} - m_V^2 \sum_k \frac{\Theta(2\omega_k - E_{\max})}{\omega_k^3} \right], \tag{108a}$$

$$\lambda_2 = -\frac{3\lambda^2}{16\pi R} \sum_k \frac{\Theta(2\omega_k - E_{\max})}{\omega_k^3}. \tag{108b}$$

The frequencies $\omega_k$ are computed using the 'quantization mass' $m_Q$, while $m_V$ is defined in Eq. (35) and evaluated in Eq. (56). The sums in $m_{V2}^2$ and $\lambda_2$ are computed numerically by summing over $k$ up to $k_{\text{UV}} = 1000$, which guarantees the inclusion of single particle states with $E \gg E_{\max}$. According to the power counting discussed in §6, the error without the terms

$\lambda_2$ and $m_{V2}^2$ should be $O(1/E_{\max}^2)$, and including the terms $\lambda_2$ and $m_{V2}^2$ should reduce the error to $O(1/E_{\max}^3)$.

Most of the calculations we perform use the reference values

$$1 = \sqrt{\frac{\lambda}{4\pi}} = m_{\text{NO}} = \frac{2\pi R}{10}\,, \tag{109}$$

in arbitrary units. The dimensions can always be restored by inserting factors of $R$ (for example) using dimensional analysis.

For the numerical truncation, we work in a Fock basis of states defined by creation operators Eq. (37):

$$|\{n\}\rangle = \left(\prod_k \frac{1}{\sqrt{n_k!}} \left(a_k^\dagger\right)^{n_k}\right)|0\rangle\,, \tag{110}$$

where $n_k$ is the number of particles with momentum $k$. The states are labeled by their eigenvalues with respect to the free Hamiltonian $H_0$ in Eq. (33) and the spatial momentum $P$:

$$H_0|\{n\}\rangle = \sum_k n_k \omega_k |\{n\}\rangle\,, \tag{111a}$$

$$P|\{n\}\rangle = \sum_k n_k k |\{n\}\rangle\,. \tag{111b}$$

In addition, the theory has a parity symmetry

$$\phi(x,t) \mapsto \phi(-x,t) \tag{112}$$

and a $\mathbb{Z}_2$ symmetry

$$\phi(x,t) \mapsto -\phi(x,t)\,. \tag{113}$$

The $\mathbb{Z}_2$ symmetry acts on our basis states as

$$Z|\{n\}\rangle = (-1)^{\sum_k n_k}|\{n\}\rangle\,. \tag{114}$$

We will present the results for states with $P = 0$, and quote values of the $\mathbb{Z}_2$ charge for excited states. For simplicity, we do not keep track of parity in this work.

The calculations were performed using very modest computing resources, namely a laptop computer running `Python` with matrix diagonalization performed using the package `scipy.sparse.linalg`.[11] The largest cutoff explored was $E_{\max} = 27$, corresponding to $\sim 9 \times 10^5$ states in the truncated Hilbert space for $m_{\text{NO}} = 1$ (see §4.3). The most computationally intensive step is the construction of the Hamiltonian matrix, which grows in size exponentially as $E_{\max}$ is increased, see Fig. 1.

We first present results that show the convergence of our method as a function of $E_{\max}$. In Figs. 2 and 3, we present numerical results for the ground state excitation energies $\Delta E_n = E_n - E_0$, where $E_0$ is the ground state energy, and $E_n$ is the energy of the $n^{\text{th}}$ excited state. When separating states into the $\mathbb{Z}_2$ even and odd bases, we label the energy levels as $\Delta E_n^\pm$. We compare the 'raw' theory, obtained by the approximation $H_{\text{eff}} = H + V$, and the improved theory, which includes the $O(V^2)$ corrections computed in §5. We fit the results to

$$\Delta E_n(E_{\max}) = \Delta E_n^{(\infty)} + \frac{C_n}{(E_{\max})^\alpha}\,, \tag{115}$$

$$m_{\mathrm{NO}} = 1, \ 2\pi R = 10$$

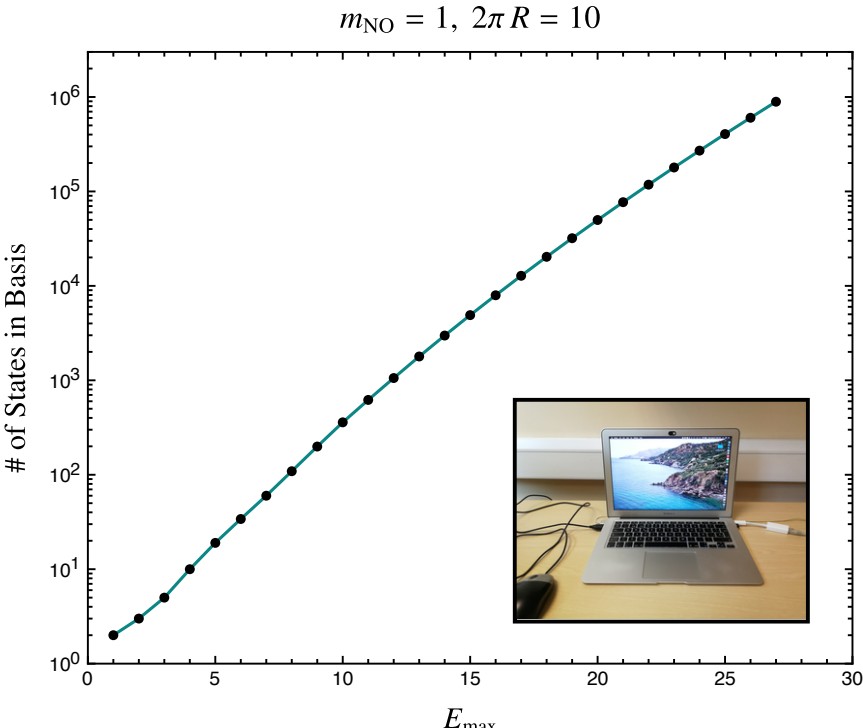

**Fig. 1.** Number of states in the basis for the $H$ matrix computed on a laptop [inset] in this work. The highest point in the plot corresponds to $E_{\max} = 27$ with $\sim 9 \times 10^5$ states.

where $\alpha = 2$ for the raw calculation, and $\alpha = 3$ for the improved theory. As predicted from the general power counting arguments presented above, the error in the raw result scales as $1/E_{\max}^2$, whereas the improved result has an error that scales as $1/E_{\max}^3$. This is a very important test that shows that our method is working as expected.

Figs. 4 and 5 show that the excitation energy $\Delta E_n$ computed in the improved theory continues to scale as $1/E_{\max}^3$ for excited states. The results are shown for states with $Z = \pm 1$, where $Z$ is the $\mathbb{Z}_2$ charge of the states (see Eq. (114)). This is encouraging for future work on extracting precision predictions for physical quantities using our method.

In Fig. 6 we present results for the first few excited states as a function of the coupling $\lambda$. The results are computed for $E_{\max} = 27$, the largest value used in our computations. We see that the eigenvalues change significantly over the range of $\lambda$ considered, a further indication that we are exploring strong coupling. It is interesting to compare the results in Fig. 6 with theoretical expectations. It is known that the theory has a second-order phase transition in the Ising universality class, where the $\mathbb{Z}_2$ symmetry is spontaneously broken for $\lambda$ larger than some critical value. In finite volume, the signal of $\mathbb{Z}_2$ symmetry breaking is that the states of the system become degenerate in pairs with $Z = \pm$. In Fig. 6 we see that this is happening at large $\lambda$ for the ground state, but not for the excited states. On the other hand, the convergence is not good at values of $\lambda$ far above the critical point. We see that the first few eigenvalues of the spectrum line up with values of the operator dimensions of the 2D Ising model within the expected range of critical coupling [6, 10, 14, 30–35]. It would be interesting to attempt to extract an accurate value for the critical coupling using our methods taking into account finite volume corrections, and compare to the results of other methods [30–35]. We leave this for the future.

---

[11]The code is available at github.com/rahoutz/hamiltonian-truncation.

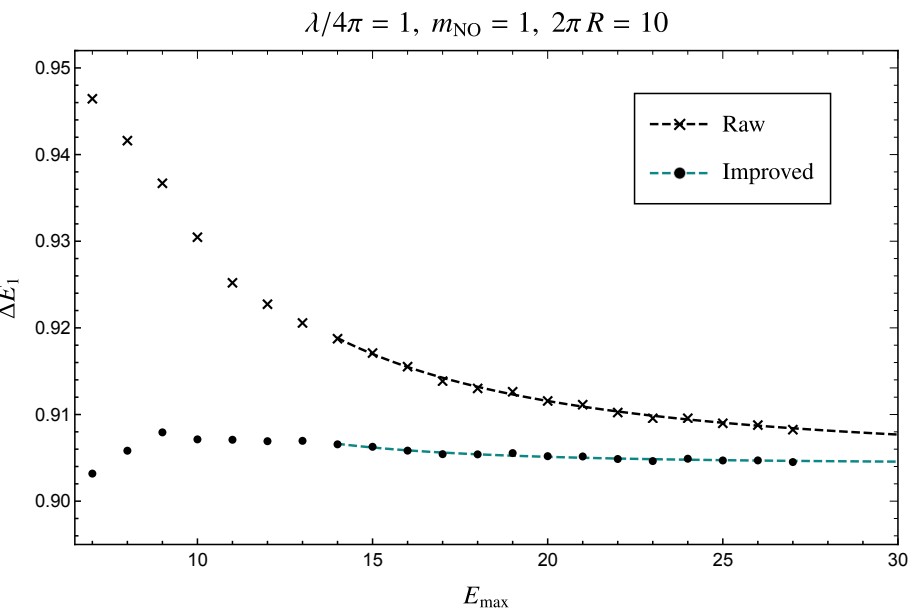

**Fig. 2.** The ground state excitation energy $\Delta E_1$ as a function of the (dimensionless) energy cutoff $E_{\max}$ for both the raw truncated [crosses] and improved [dots] results. For $E_{\max} \geq 14$, the ground state excitation energy of the raw and improved theories are fit to $1/E_{\max}^2$ and $1/E_{\max}^3$, respectively.

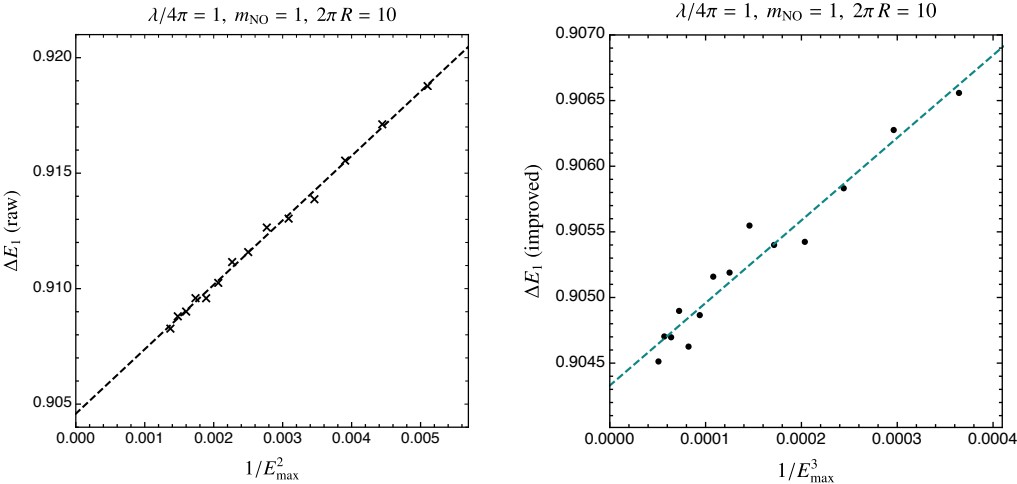

**Fig. 3.** We highlight the power-law scaling by zooming in on the flat $E_{\max} \geq 14$ tail and plotting the ground state excitation energy $\Delta E_1$ as a function of $1/E_{\max}^2$ for the raw truncated theory [left] and as a function of $1/E_{\max}^3$ for the improved theory [right]. The asymptotic values of $\Delta E_1^\infty$ extracted from the fits are in close agreement: $\Delta E_1^\infty = 0.9046$ (raw) and $\Delta E_1^\infty = 0.9043$ (improved).

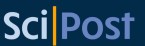

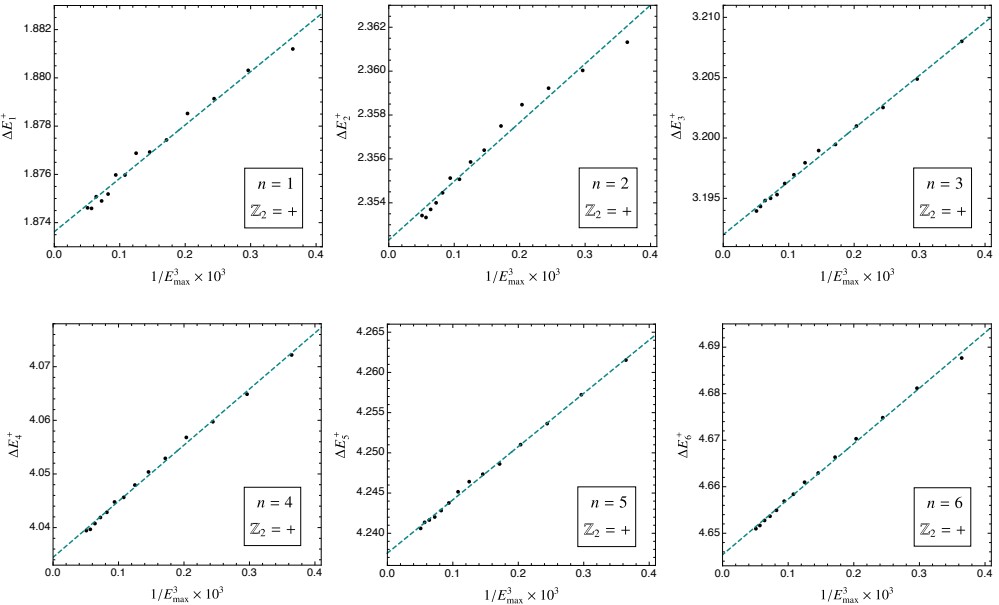

**Fig. 4.** The excitation energy $\Delta E_n$ of the first six $\mathbb{Z}_2$-even excited states in the improved theory as a function of $1/E_{\text{max}}^3$, including the best fit line. In all panels, $\lambda/4\pi = 1$, $m_{\text{NO}} = 1$, $2\pi R = 10$.

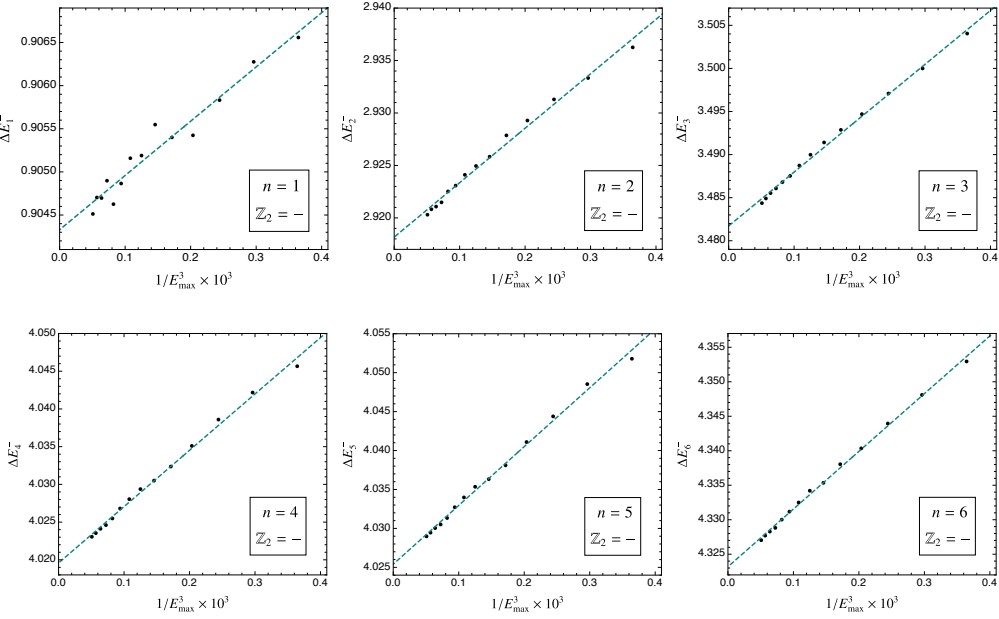

**Fig. 5.** The excitation energy $\Delta E_n$ of the first six $\mathbb{Z}_2$-odd excited states in the improved theory as a function of $1/E_{\text{max}}^3$, including the best fit line. In all panels, $\lambda/4\pi = 1$, $m_{\text{NO}} = 1$, $2\pi R = 10$.

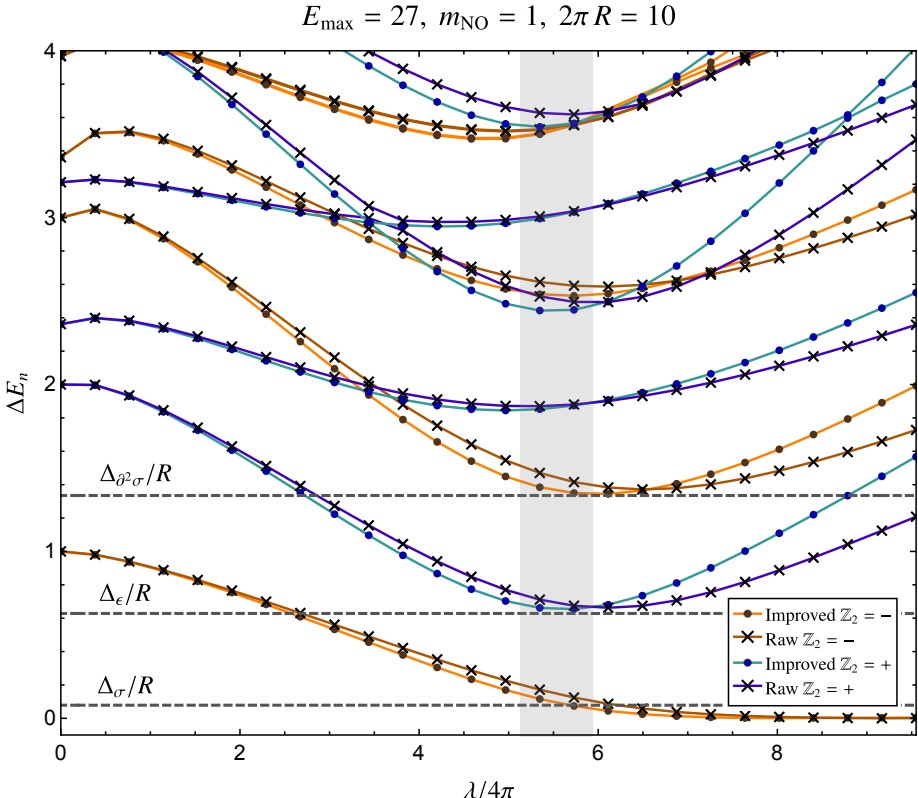

**Fig. 6.** The excitation energy spectra as a function of the $\phi^4$ coupling $\lambda/4\pi$. The states belonging to the $\mathbb{Z}_2$-even (-odd) basis are shown in blue (orange). The raw truncated theory is plotted in dark blue (dark orange) using cross markers, while the improved theory is plotted light blue (orange) using dot markers. For clarity, lines are drawn to connect the data points within each excited state. The dashed horizontal lines correspond to the known theoretical values for the operator dimensions of the 2D Ising model, and the grey band covers the predicted range of the critical coupling.

In Figs. 7 and 8, we present results on the convergence of $\Delta E_1$ for $\lambda/4\pi = 2, 4, 6, 8$ in both the $\mathbb{Z}_2$-odd and -even bases. The errors generally continue to scale as $1/E_{\max}^3$ for larger values of $\lambda$. Numerical noise spoils the convergence of $\Delta E_1^-$ for $\lambda/4\pi = 8$ (Fig. 7), corresponding to the region where the first $\mathbb{Z}_2$-odd excited state and the vacuum are becoming degenerate.[12]

Finally, in Fig. 9 we show how our results change as a function of the quantization mass $m_Q$. In our approach, $m_Q$ is an arbitrary variational parameter that can be used to optimize the convergence. It is defined by taking the particle energies in the Fock space to be given by $\omega_k = \sqrt{(k/R)^2 + m_Q^2}$ (see §4.3). To provide a 'fair' comparison, we vary $m_Q$ keeping the number of states in the truncation fixed, since this is what determines the required computational resources. Our previous results have been presented with $m_Q = m_{NO}$, and in Fig. 9 we compare these results with $m_Q = 2m_{NO}$ and $m_Q = m_{NO}/2$ . We see that the convergence appears to be best for $m_Q = m_{NO}$. Physically, we expect that the optimal value of $m_Q$ is to take it equal to the physical mass scale of the theory, since then the Fock states are expected to give the best variational basis for the interacting states. It would be interesting to explore the optimal choice of $m_Q$ in more complicated situations with more than one scale, for example taking the volume to be larger to extrapolate to $R \to \infty$. We also leave this for future work.

---

[12]The `scipy.sparse.linalg` matrix diagonalization uses the implicitly restarted Arnoldi method [36,37], which may not be reliable for systems with nearly degenerate eigenvalues [38].

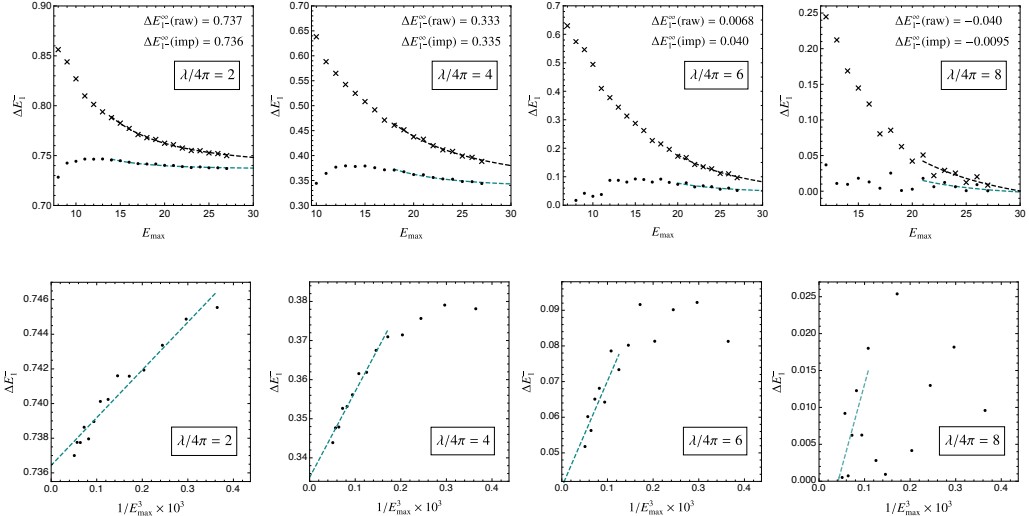

**Fig. 7.** The $\mathbb{Z}_2$-odd ground state excitation energy $\Delta E_1^-$ dependence on the energy cutoff $E_{\max}$ for $\lambda/4\pi = 2, 4, 6, 8$. In all panels, $m_{\mathrm{NO}} = 1$ and $2\pi R = 10$. The high $E_{\max}$ tails of $\Delta E_1$ for the raw and improved theories are fit to $1/E_{\max}^2$ and $1/E_{\max}^3$, respectively. The convergence is spoiled by numerical noise for larger values of $\lambda$.

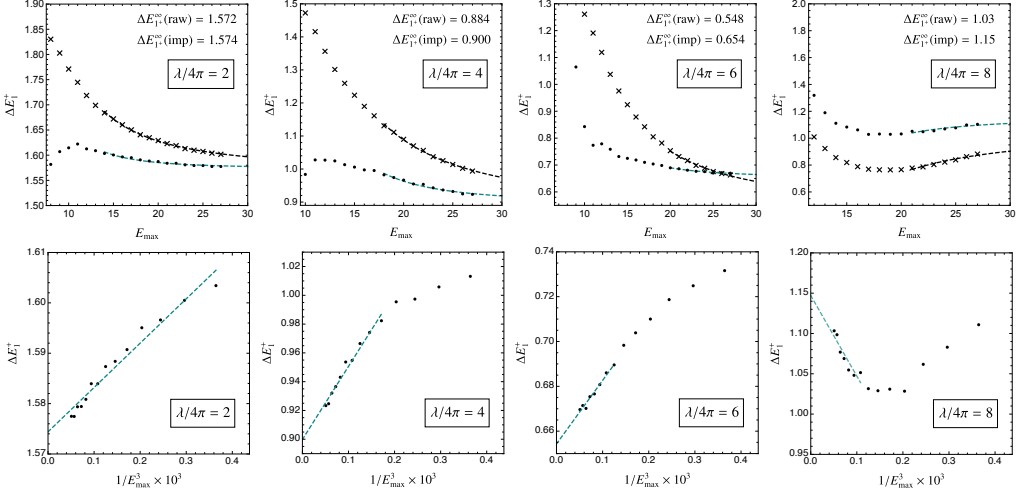

**Fig. 8.** The $\mathbb{Z}_2$-even ground state excitation energy $\Delta E_1^+$ dependence on the energy cutoff $E_{\max}$ for $\lambda/4\pi = 2, 4, 6, 8$. In all panels, $m_{\mathrm{NO}} = 1$ and $2\pi R = 10$. The high $E_{\max}$ tails of $\Delta E_1^+$ for the raw and improved theories are fit to $1/E_{\max}^2$ and $1/E_{\max}^3$, respectively. The agreement between the asymptotic values of the raw and improved theories degrades for larger couplings. In the bottom panels, we see that the improved result maintains $1/E_{\max}^3$ scaling to good approximation for strong couplings.

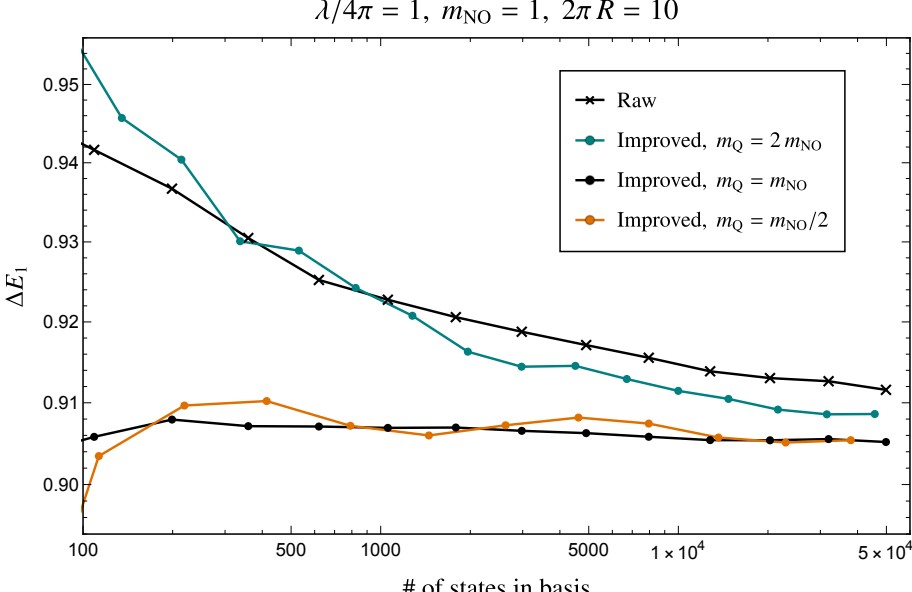

**Fig. 9.** The ground state excitation energy $\Delta E_1$ of the raw theory [crosses] and the improved theory [dots] for $m_Q = 2m_{NO}$, $m_{NO}$, $m_{NO}/2$. Here $\Delta E_1$ is plotted against the size of the basis to better compare the convergence given finite computational resources. The data points are connected by lines to guide the eye. This figure includes data points for theories with up to $5 \times 10^4$ states in the basis. With this upper bound, we probe theories with $E_{max}$ up to $E_{max} = \{16, 20, 26\}$ for $m_Q = \{1/2, 1, 2\} \times m_{NO}$.

# 8 Comparison with Previous Work

In this section, we compare our approach to other approaches to improving the accuracy of Hamiltonian truncation in quantum field theory.

## 8.1 Extending the Basis

The first attempt to improve the predictions of Hamiltonian truncation for quantum field theory appears to be Ref. [39]. In this approach, the truncation to the space $\mathcal{H}_{eff}$ spanned by $H_0$ eigenstates with eigenvalue $E \leq E_{max}$ is the first step in an iterative procedure that aims to improve the convergence. At each step, one has a finite-dimensional subspace $\mathcal{H}_{eff}^{(n)}$ that is updated by adding and removing vectors to improve the convergence. Basis vectors are removed if their overlap with the (truncated) eigenvectors is smaller than some chosen value, and a stochastic algorithm is used to identify basis states with larger overlap.

This method is entirely numerical, and does not exploit the fact that the theory is weakly coupled in the UV. Also, it may be difficult to extend this method to theories with UV divergences. In such theories, the total contribution of modes with high energy grows with the energy, due to the exponential increase of the number of modes. The contribution of individual states above the truncation cutoff is exponentially small, but their total contribution is infinite.

## 8.2 Renormalization Group Equation

Another approach to improving Hamiltonian truncation was proposed in the context of 2D conformal theories perturbed by relevant operators in Refs. [40, 41]. The idea is to define

a renormalization group (RG) equation for the coupling constants of the theory, namely the coefficients of the relevant operators. (In the case of 2D $\lambda\phi^4$ theory, these are the coupling $\lambda$ and the mass $m^2$.) The RG equation is determined by requiring that certain correlation functions are independent of the truncation cutoff. Because the perturbations are relevant, the RG equations can be computed in a perturbative expansion in the couplings.

As in our paper, their approach takes advantage of the weak coupling of the theory in the UV, and computes a correction to the truncated Hamiltonian by a perturbative matching calculation. Our work can be viewed as an extension of this approach that systematically includes additional terms in the truncated Hamiltonian.

## 8.3 Exact Effective Hamiltonian

This approach was introduced in Refs. [5,6], and further developed in Ref. [14]. The starting point is the observation that one can rewrite the exact eigenvalue equation Eq. (5) in terms of an 'exact effective Hamiltonian' (our terminology) acting on the low-energy subspace $\mathcal{H}_{\text{eff}}$:

$$H_{\text{exact}}(\mathcal{E})|\mathcal{E}\rangle_{\text{eff}} = \mathcal{E}|\mathcal{E}\rangle_{\text{eff}}, \tag{116}$$

where $|\mathcal{E}\rangle_{\text{eff}} \in \mathcal{H}_{\text{eff}}$. The exact effective Hamiltonian is given by

$$H_{\text{exact}}(\mathcal{E}) = H_0 + V + \Delta H_{\text{exact}}(\mathcal{E}), \tag{117}$$

with

$$\langle f|\Delta H_{\text{exact}}(\mathcal{E})|i\rangle = -\langle f|VP_>(H_0 + P_>VP_> - \mathcal{E})^{-1}P_>V|i\rangle, \tag{118}$$

where $|i\rangle, |f\rangle \in \mathcal{H}_{\text{eff}}$ and $P_>$ is the projector onto states with $E > E_{\text{max}}$.

Note that $\Delta H_{\text{exact}}(\mathcal{E})$ depends on the eigenvalue one is trying to compute. In principle, one can vary $\mathcal{E}$ iteratively to converge to the correct eigenvalue, but this is numerically expensive. Instead Refs. [5,6] diagonalize $H_{\text{exact}}(\mathcal{E}_*)$, where $\mathcal{E}_*$ is a reference value chosen to be close to the low-lying eigenvalues one is computing. Next, these authors evaluate the right-hand side of Eq. (118) using an approximation that is valid for $E_\alpha \gg E_{\text{max}}$. The resulting leading corrections to the effective Hamiltonian are local, like the leading $O(V^2)$ corrections we computed in §5. Numerical diagonalization of this approximation to $H_{\text{exact}}(\mathcal{E}_*)$ for 2D $\lambda\phi^4$ theory shows significantly improved convergence, but the errors are larger than the $O(1/E_{\text{max}}^3)$ errors we obtain from our method.

In Ref. [14], an extension of this method was found that does give $O(1/E_{\text{max}}^3)$ errors for 2D $\lambda\phi^4$ theory. This was accomplished by adding judiciously chosen additional states above the cutoff (called 'tail states' by the authors). Our approach obtains $O(1/E_{\text{max}}^3)$ errors without enlarging the basis.

## 8.4 Counterterms in Theories with UV Divergences

Refs. [21,23] were the first to 'break the UV divergence barrier' and successfully apply Hamiltonian truncation to a theory with UV divergences, namely $\lambda\phi^4$ theory in 3D. In such theories, the 'raw' truncation approximation $H_{\text{eff}} = H_0 + V$ does not converge as $E_{\text{max}}$ is increased.

Ref. [21] analyzed perturbation theory for the energy eigenvalues and identified UV divergent contributions arising from sums over states with energy above $E_{\text{max}}$. They then defined a renormalized effective Hamiltonian by subtracting these contributions, and demonstrated that numerical diagonalization of this Hamiltonian gives sensible results.

Ref. [23] analyzed the theory using a different truncation applied to the lightcone Hamiltonian. They identified contributions with unphysical non-local cutoff dependence, which they



interpreted as artifacts of their truncation and therefore subtracted. Numerical diagonalization of this Hamiltonian gives also gives sensible results.

As we discussed in §2.1, matching the energy eigenvalues is not sufficient to completely determine the effective Hamiltonian. Therefore, our work can be viewed as a systematic extension of the methods of Refs. [21, 23] that allows further improvement of the results.

# 9 Conclusions

In this paper, we have formulated Hamiltonian truncation as an effective field theory (HTET) with a cutoff on the total energy $E_{\max}$. This allows us to perform matching calculations to compute corrections to the effective Hamiltonian as a systematic expansion in powers of $1/E_{\max}$ using a diagrammatic approach. We demonstrated this method by applying it to $\lambda\phi^4$ theory in 2D. We computed the leading corrections to the 'raw' Hamiltonian truncation in this model, and carried out numerical tests of the improved truncation. The calculation involves 2- and 3-loop diagrams with overlapping UV/IR dominated regions, and we demonstrated that the results are compatible with the separation of scales, as we expect in effective field theory. We also performed calculations up to 2 loops in 3D $\lambda\phi^4$ theory to illustrate the application of our methods in theories with nontrivial UV divergences. Our numerical studies of the 2D theory showed that the corrected theory has an error that scales as $1/E_{\max}^3$, the theoretically expected scaling from the power counting of the effective theory. These results are an indication that our method is working as expected.

There are many directions to explore in future work. We begin by mentioning several projects that are straightforward applications of the formalism developed in this paper. One is to compute the next order corrections to 2D $\lambda\phi^4$ theory, where the effective Hamiltonian is neither local nor Hermitian. This calculation would give a nontrivial test of the power counting we proposed for the effective theory. At this order, the truncation error is expected to scale as $1/E_{\max}^4$, and we expect to get predictions accurate to 5 significant digits. Another project is to finish the calculations for the leading corrections to 3D $\lambda\phi^4$ theory, together with a numerical analysis to demonstrate the method in a theory in higher dimensions with nontrivial UV divergences. A complementary direction is to use Hamiltonian truncation to extract precision predictions for physical observables, such as the critical coupling, or physical masses and scattering amplitudes away from criticality. This would require a careful treatment of finite volume effects and extrapolation errors.

We also suggest some more speculative directions. The ultimate goal of Hamiltonian truncation is to improve on existing numerical methods to study quantum field theories, such as lattice Monte Carlo. Currently, we are very far from this goal. For example, there is no general method to treat gauge theories in Hamiltonian truncation, and there has been no successful Hamiltonian truncation calculation in 4 spacetime dimensions. Hamiltonian truncation suffers from the fact that the number of states grows exponentially with $E_{\max}$, with an exponent that grows with the spacetime dimension. Nonetheless, we are optimistic. For example, quantum computers may eventually be able to overcome the exponential growth in computational complexity in Hamiltonian truncation. Until then, there are interesting 2D and 3D models that can be studied using Hamiltonian truncation, including theories that are not accessible to lattice techniques, such as theories with chiral fermions or supersymmetry. It is our hope that the conceptual power of understanding Hamiltonian truncation in the language of effective field theory will help us make progress on all these outstanding problems.

## Acknowledgments

We thank Joan Elias-Miró, Marat Freytsis, Brian Henning, Emanuel Katz, and Matthew Walters for useful discussions, and Slava Rychkov for comments on the draft. The referees contributed a number of useful comments: we thank Balt van Rees, Slava Rychkov, and an anonymous referee for their input. T. Cohen is supported by the DOE under grant DE-SC-0011640. K. Farnsworth is supported by the Simons Foundation under award 658908. R. Houtz is supported by the STFC under grant ST/P001246/1. M. A. Luty is supported by the DOE under grant DE-SC-0009999.

# Appendices

# A  Renormalization and Matching in 3D $\lambda \phi^4$ Theory

In this section, we present some renormalization and matching calculations for 3D $\lambda \phi^4$ theory. Although this theory is also super-renormalizable, its UV divergence structure is more complicated than in 2D $\lambda \phi^4$ theory. The calculations in this section illustrate that separation of scales continues to work in theories with UV divergences.

## A.1  Power Counting

The mass dimensions of the field and couplings in 3D are

$$[\phi] = \frac{1}{2}, \qquad [\lambda] = 1. \tag{1}$$

The most general parametric form of the effective Hamiltonian allowed by the power counting presented in §6 is

$$
\begin{aligned}
H_2 + H_3 + \cdots \sim \int \mathrm{d}^2 x \bigg\{ & \lambda^2 \Big( E_{\max} \cdot \mathbb{1} + H_0 \ln E_{\max} + \frac{H_0^2}{E_{\max}} + \cdots \Big) \\
& + \lambda^2 \phi^2 \Big( \ln E_{\max} + \frac{H_0}{E_{\max}} + \cdots \Big) \\
& + \lambda^2 \big( \phi^4 + \mathrm{i} \phi \dot{\phi} \big) \Big( \frac{1}{E_{\max}} + \frac{H_0}{E_{\max}^2} + \cdots \Big) \\
& + \lambda^3 \Big( \ln E_{\max} \cdot \mathbb{1} + \frac{H_0}{E_{\max}} + \cdots \Big) \\
& + \lambda^3 \phi^2 \Big( \frac{1}{E_{\max}} + \frac{H_0}{E_{\max}^2} + \cdots \Big) \cdots \bigg\}.
\end{aligned}
\tag{2}
$$

Note that the power counting suggests that the coefficients of $\lambda^2 H_0$ and $\lambda^2 \phi^2$ increase logarithmically with $E_{\max}$. We therefore expect that the 'raw' truncation $H_{\text{eff}} \simeq H_0 + H_1$ does not converge as $E_{\max}$ is increased as opposed to the 2D theory, but including the leading $O(\lambda^2)$ terms that grow with $E_{\max}$ will give a truncation with errors of order $1/E_{\max}$. Refs. [21,23] successfully carried out numerical studies of this theory using a different renormalization method; we compare our approach with theirs in §8.4.

## A.2 Renormalization of the Fundamental Theory

We write the bare Lagrangian as

$$\mathcal{L} = \frac{1}{2}(\partial \phi)^2 - \frac{1}{2}m_0^2 - \frac{\lambda}{4!}\phi^4. \tag{3}$$

We again regulate the fundamental theory by imposing a momentum space cutoff $\Lambda$:

$$\sum_{\boldsymbol{k}} \quad \rightarrow \quad \sum_{\boldsymbol{k}}^{\Lambda} \equiv \sum_{|\boldsymbol{k}| \le \Lambda R}, \tag{4}$$

where the sum is over $\boldsymbol{k} \in \mathbb{Z}^2$. This cutoff breaks Lorentz invariance, and therefore we must allow counterterms that break Lorentz invariance. However, the cutoff is local, so the counterterms are given by local terms:

$$\delta \mathcal{L} \sim \lambda \phi^2 \Lambda + \lambda^2 \phi^2 \ln \Lambda + \text{finite}, \tag{5}$$

where we have omitted cosmological constant terms that are independent of $\phi$. We see that there is no need for wavefunction or coupling constant renormalization, as in the 2D theory. Since we are only interested in differences of energy eigenvalues, we ignore the vacuum energy divergences and focus on the terms proportional to $\phi^2$.

The Hamiltonian is $H_0 + V$, where

$$H_0 = \sum_{\boldsymbol{k}}^{\Lambda} \omega_{\boldsymbol{k}} a_{\boldsymbol{k}}^\dagger a_{\boldsymbol{k}}, \qquad \omega_{\boldsymbol{k}} = \sqrt{|\boldsymbol{k}|^2/R^2 + m_Q^2}, \tag{6a}$$

$$V = \int \mathrm{d}^2 x \left[ \frac{1}{2}(m_V^2 + \delta m_V^2) : \phi^2 : + \frac{\lambda}{4!} : \phi^4 : \right], \tag{6b}$$

where

$$m_V^2 + \delta m_V^2 = m_0^2 - m_Q^2 + \frac{\lambda}{16\pi^2 R^2} \sum_{\boldsymbol{k}}^{\Lambda} \frac{1}{\omega_{\boldsymbol{k}}}. \tag{7}$$

Here $\delta m_V^2$ is a counterterm that will be used to absorb the $\lambda^2 \phi^2 \ln \Lambda$ divergence. The diagrammatic rules for the vertices are then

$$\begin{array}{c} k_3 \diagdown \diagup k_1 \\ \times \\ k_4 \diagup \diagdown k_2 \end{array} = \frac{\lambda}{(2\pi R)^2} \delta_{\boldsymbol{k}_1 + \cdots + \boldsymbol{k}_4}, \tag{8a}$$

$$k_2 \text{——}\bullet\text{——} k_1 = m_V^2 \delta_{\boldsymbol{k}_1 \boldsymbol{k}_2}, \tag{8b}$$

$$k_2 \text{——}\blacksquare\text{——} k_1 = \delta m_V^2 \delta_{\boldsymbol{k}_1 \boldsymbol{k}_2}. \tag{8c}$$

As in the 2D theory, we also define a running renormalized mass by subtracting the contribution of modes with $k > \mu/R$:

$$m_R^2(\mu) + \delta m^2(\mu) = m_0^2 + \frac{\lambda}{16\pi^2 R^2} \sum_{|\boldsymbol{k}| > \mu R}^{\Lambda} \frac{1}{\omega_{\boldsymbol{k}}}, \tag{9}$$

where the $O(\lambda^2)$ $\delta m^2$ is allowed to depend on the renormalization scale $\mu$. We will see that separation of scales is manifest in terms of the renormalized coupling $m_R^2$. Note that if we choose the $O(\lambda^2)$ counterterms to satisfy $\delta m_V^2 = \delta m^2(\mu = 0)$, we have

$$m_V^2 = m_R^2(\mu) - m_Q^2 + \frac{\lambda}{16\pi^2 R^2} \sum_{|\mathbf{k}| \le \mu R} \frac{1}{\omega_k}, \tag{10}$$

just as in the 2D theory.

The $O(\lambda^2)$ diagrams that we must compute to renormalize the $\phi^2$ term are given in Eq. (77). The power counting argument above shows that these contributions are independent of the external momenta and energies, so we can compute them in the local approximation. We obtain[13]

$$\simeq -\frac{\lambda^2}{768\pi^4 R^4} \int d^2x \, \langle f| {:}\phi^2{:} |i\rangle \times \sum_{\mathbf{k}_{1,2,3}}^{\Lambda} \frac{\delta_{123,0}}{\omega_1 \omega_2 \omega_3 (\omega_1 + \omega_2 + \omega_3)}. \tag{11}$$

This diagram contains mixed UV/IR regions where one of the 3 momenta is much smaller than the others. In this region, we have

$$\sum_{\mathbf{k}_{1,2,3}}^{\Lambda} \frac{\delta_{123,0}}{\omega_1 \omega_2 \omega_3 (\omega_1 + \omega_2 + \omega_3)} \simeq 3 \times \sum_{|\mathbf{k}| \gg \mu R} \frac{1}{2\omega_k^3} \sum_{|\mathbf{k}'| \ll \mu R} \frac{1}{\omega_{k'}} + \cdots, \tag{12}$$

which is finite as $\Lambda \to \infty$. Therefore, there are no overlapping UV divergences in this diagram. The remaining diagrams are given by

$$\simeq -\frac{\lambda m_V^2}{64\pi^2 R^2} \int d^2x \, \langle f| {:}\phi^2{:} |i\rangle \times \sum_{\mathbf{k}}^{\Lambda} \frac{1}{\omega_k^3}, \tag{13}$$

which is also finite.

We conclude that the only UV divergent $\phi^2$ term comes from the region where all 3 momenta in Eq. (11) are large. We therefore define the counterterm $\delta m^2(\mu)$ to cancel the contribution of this region:

$$\delta m^2(\mu) = \frac{\lambda^2}{768\pi^4 R^4} \sum_{|\mathbf{k}_{1,2,3}| > \mu R}^{\Lambda} \frac{\delta_{123,0}}{\omega_1 \omega_2 \omega_3 (\omega_1 + \omega_2 + \omega_3)} \sim \ln(\Lambda/\mu). \tag{14}$$

Note we use the renormalization scale $\mu$ to define the counterterm. At $O(\lambda^2)$, we also have contributions to the $\phi^4$ vertex of the form (in the local approximation)

$$\simeq -\frac{\lambda^2}{256\pi^2 R^2} \int d^2x \, \langle f| {:}\phi^4{:} |i\rangle \times \sum_{\mathbf{k}}^{\Lambda} \frac{1}{\omega_k^3}, \tag{15}$$

which is finite.

---

[13]The results can be read off from the 2D results for the same diagrams with the replacements $\lambda \to \lambda/2\pi R$, and $\sum_k \to \sum_{\mathbf{k}}$.

### A.3 Matching and Separation of Scales at $O(V)$

We now turn to the matching calculation of the effective Hamiltonian, starting at $O(V)$. Using the same notation as the 2D theory, we write

$$H_1 = \int d^2 x \left[ \frac{1}{2} m_1^2 \phi^2 + \frac{\lambda_1}{4!} \phi^4 \right] = \int d^2 x \left[ \frac{1}{2} m_{V1}^2 : \phi^2 : + \frac{\lambda_{\text{eff}}}{4!} : \phi^4 : \right], \tag{16}$$

where

$$m_{V1}^2 = m_1^2 + \frac{\lambda_1}{16\pi^2 R^2} \sum_{|\mathbf{k}| \le k_{\max}} \frac{1}{\omega_{\mathbf{k}}} \tag{17}$$

and $\omega_{\mathbf{k}_{\max}} = E_{\max}$. Matching the coefficients of the normal-ordered operators then gives

$$\lambda_{\text{eff}} = \lambda, \qquad m_{V1}^2 = m_V^2. \tag{18}$$

As in 2D, $m_{V1}^2$ is independent of $\mu$, and the coefficient of $\phi^2$ in $H_{\text{eff}}$ at $O(\lambda)$ is given by

$$m_Q^2 + m_1^2 = m_R^2 (\mu = k_{\max}/R). \tag{19}$$

We see that separation of scales is manifest in terms of $m_R^2 (\mu \sim E_{\max})$, just as in the 2D theory.

### A.4 Matching and Separation of Scales at $O(V^2)$

Using the local approximation, the 2-loop diagrams are given by

$$\simeq -\frac{\lambda^2}{768\pi^4 R^4} \int d^2 x \langle f | : \phi^2 : | i \rangle \times \sum_{\mathbf{k}_{1,2,3}}^{\Lambda} \delta_{123} \frac{\Theta(\omega_1 + \omega_2 + \omega_3 - E_{\max})}{\omega_1 \omega_2 \omega_3 (\omega_1 + \omega_2 + \omega_3)}. \tag{20}$$

This has a UV divergence that is canceled by the counterterm Eq. (14) in the fundamental theory. The 1-loop diagram contributions to $\phi^2$ are given by

$$\simeq -\frac{\lambda m_V^2}{64\pi^2 R^2} \int d^2 x \langle f | : \phi^2 : | i \rangle \sum_{\mathbf{k}}^{\Lambda} \frac{\Theta(2\omega_{\mathbf{k}} - E_{\max})}{\omega_{\mathbf{k}}^3}. \tag{21}$$

Finally the 1-loop contributions to $\phi^4$ are given by

$$\simeq -\frac{\lambda^2}{256\pi^2 R^2} \int d^2 x \langle f | : \phi^4 : | i \rangle \times \sum_{\mathbf{k}}^{\Lambda} \frac{\Theta(2\omega_{\mathbf{k}} - E_{\max})}{\omega_{\mathbf{k}}^3}. \tag{22}$$

The $\phi^2$ term (not normal-ordered) in the effective Hamiltonian is then given by

$$H_2 = \int d^2 x \frac{1}{2} m_2^2 \phi^2 + \cdots, \tag{23}$$

with

$$m_2^2 = -\frac{\lambda^2}{384\pi^4 R^4} \sum_{k_{1,2,3}}^{\Lambda} \delta_{123,0} \frac{\Theta(\omega_1 + \omega_2 + \omega_3 - E_{\max}) - \prod_{i=1}^{3} \Theta(|k_i| - \mu R)}{\omega_1 \omega_2 \omega_3 (\omega_1 + \omega_2 + \omega_3)}$$

$$- \frac{\lambda}{32\pi^2 R^2} m_V^2 \sum_{k}^{\Lambda} \frac{\Theta(2\omega_k - E_{\max})}{\omega_k^3}$$

$$+ \frac{3\lambda^2}{512\pi^4 R^4} \sum_{k}^{\Lambda} \frac{\Theta(2\omega_k - E_{\max})}{\omega_k^3} \sum_{|k'| \le k_{\max}} \frac{1}{\omega_{k'}}. \tag{24}$$

The second step function in the first sum comes from the counterterm Eq. (14) and the third line in the sum is from un-normal-ordering the $\phi^4$ correction Eq. (22). This is UV finite, and can be evaluated numerically. We can also see the separation of scales if we choose the renormalization scale $\mu \sim E_{\max}$ in the expression of $m_V^2$ Eq. (10). Focusing on the terms with mixed UV/IR behavior, we have

$$m_2^2 \simeq -\frac{\lambda^2}{128\pi^4 R^4} \underbrace{\left(\frac{1}{2} + \frac{1}{4} - \frac{3}{4}\right)}_{=0} \sum_{|k'| \ll k_{\max}} \frac{1}{\omega_{k'}} \sum_{k}^{\Lambda} \frac{\Theta(2\omega_k - E_{\max})}{\omega_k^3} + \cdots \tag{25}$$

and we see exactly the same cancellation as in the 2D case Eq. (87).

## B A Hermitian Effective Hamiltonian?

The effective Hamiltonian $H_{\text{eff}}$ defined in this paper is non-Hermitian. In this appendix, we look for a similarity transformation

$$H'_{\text{eff}} = G H_{\text{eff}} G^{-1}, \tag{26}$$

so that $H'_{\text{eff}}$ is Hermitian. We will consider this for the 2D $\lambda \phi^4$ theory, where the leading non-Hermitian terms in the $1/E_{\max}$ expansion involve $\phi^2 H_0$ and $\phi^4 H_0$ (see Eq. (102)). Writing $\phi^n H_0 = \frac{1}{2}\{\phi^n, H_0\} + \frac{1}{2}[\phi^n, H_0]$, we have

$$H_{\text{eff}} = \text{Hermitian} + \frac{\lambda^2}{E_{\max}^3} \int dx \left( c_2 [\phi^2, H_0] + c_4 [\phi^4, H_0] \right) + O(1/E_{\max}^4). \tag{27}$$

We therefore define $G$ perturbatively

$$G = 1 + X + O(X^2), \tag{28}$$

which gives

$$H'_{\text{eff}} = H_{\text{eff}} + [X, H_{\text{eff}}] + O(X^2). \tag{29}$$

Comparing Eqs. (27) and (29), we see that a natural choice is

$$X = \epsilon H_0. \tag{30}$$

Since

$$V = \int dx \left[ \frac{1}{2} m_V^2 \phi^2 + \frac{\lambda}{4!} \phi^4 \right] \tag{31}$$

this gives

$$[X, H_{\text{eff}}] = \epsilon \int dx \left( \frac{1}{2} m_V^2 [H_0, \phi^2] + \frac{\lambda}{4!} [H_0, \phi^4] \right) + O(\epsilon \lambda^2) + O(\epsilon^2). \tag{32}$$

Comparing with Eq. (27), we see that we can choose $\epsilon$ to cancel one of the two leading non-Hermitian terms, but not both.

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
