# Peer review of "Hamiltonian Truncation Effective Theory"

_SciPost Physics, doi:SciPost Phys. 13, 011 (2022)_

## Round 1 · Referee Report · Slava Rychkov (Referee 2) · 2022-1-7

Report

This paper aims to improve convergence of the Hamiltonian Truncation, by adding to the truncated Hamiltonian a series of judiciously chosen correction terms. The idea itself is not new, and some proposals exist in the literature. The paper proposes a new method to do it, rooted in Effective Field Theory. The proposal identifies a new quantity - called transition matrix - which should be computed in the fundamental and in the truncated theory and matched. The matching condition determines which correction terms one should add in the truncated theory.

The paper is interesting and generally well written although omitting some details. I would like to list several questions, comments, and suggestions which occurred to me when reading it. These fall in three categories: the need to explain better conceptual foundations of their method; the need to document better their computations to allow reproducibility; the need to localize better the difference from the previous work by Rychkov and Vitale. I suggest, although I do not insist, that the authors add clarifications concerning at least some of these points. The framework and the results are promising. I wish the authors good luck in pushing their program to correction terms of higher order, and to higher dimensions.

1) Numerical section 7 could benefit from extra details clarifying the meaning of "improved" results. Full improved Hamiltonian could be described in detail. At least it should be said if the matching corrections are used for states i,f with energies all the way up to the cutoff (since the previous sections compute them under the constraint E_i,E_f<<E_max).

Elsewhere in the text, it is mentioned that sums involved in the evaluating matching corrections are evaluated sometimes analytically (in 1/E_max expansion), sometimes numerically. Summarizing the procedure in section 7 could be helpful, to allow reproducibility.

2) Their matching corrections, at the order they аrе working, are very similar, although hopefully not identical, to the correction terms used by Rychkov and Vitale in "Hamiltonian Truncation Study of the φ^4 Theory in Two Dimensions", http://arxiv.org/abs/1412.3460. Although derived in that paper via an OPE argument, Eq. (3.31) can be shown to arise from local approximation to the sums very similar to the sums used in the paper under review (appendix E in http://arxiv.org/abs/1706.09929). The 1/E_max expansion of matching corrections could shed light on why correction terms of Rychkov and Vitale do not improve correction to O(1/Emax^3), while the matching corrections in the paper do so. Currently, the comparison to the prior work by Rychkov and Vitale in section 8.3 does not explain this issue. One thing to look at is \mu_{442} in (3.31) of Eq. (3.31) of Rychkov and Vitale which contains log(Lambda/m), while all matching corrections in the paper under review are argued to be purely high-energy associated and should presumably be free of such logs. A precise formula analogous to (3.31) in Rychkov and Vitale could therefore be welcome. It is derivable by methods of appendix E in http://arxiv.org/abs/1706.09929.

3) The logic on p.8 concerning the phrase "We therefore need additional relations to fully determine effective Hamiltonian" is unclear. It looks like a good effective Hamiltonian is any matrix which has the same low-energy spectrum as the full Hamiltonian. Why is it bad that section 2.1 does not determine it fully? Why can't we e.g. put all elements which are not fixed by section 2.1 to zero, or to anything else of a given order in V? Why would this be a worse definition than the one the authors come up with in Section 2.2? Put another way, is the freedom accorded by Section 2.1 perhaps just a gauge freedom, and Section 2.2 a particular way to fix that freedom, or is there something more special about the choice of Section 2.2?

3') A related remark. The authors stress that their effective Hamiltonian is not Hermitian. Could this be cured? Hamiltonians H and H' = G^{-1} H G have the same spectrum. For G unitary, H' is Hermitian if and only if H is so. However for a general invertible G we may have H non-Hermitian and H' Hermitian. It might therefore very well be possible to transform their Hamiltonian to an equivalent Hermitian one. Such a transformation would be advantageous for numerics, since Hermitian matrices are faster to diagonalize. Do the authors see any particular reason why such a transformation may not exist, or should not be used?

4) What is the physical meaning of the transition matrix introduced in section 2.2 ? Is it observable? Quantum Mechanics textbooks contain similar quantities, but not exactly this one. In Quantum Mechanics, for adiabatically turning on interactions, transition is only possible between eigenstates having the same energy, while the transition matrix in the paper has no such constraint. So it looks like their transition matrix is something else from what it is in Quantum Mechanics. They say that it is related to S-matrix, but also S-matrix conserves energy... Given that all possible quantities of quantum-mechanical origin have been presumably considered before, it would be great to identify the previous uses of such "transition matrices between states of different energies", and cite them, to orient the reader. If to the authors' knowledge it's the first time such a quantity is used in the long history of quantum mechanics, this is remarkable and should also be mentioned.

5) It would seem required by the logic of the paper that the effective Hamiltonian satisfying the conditions from section 2.2 satisfies also those from section 2.1, but this fact is neither stated nor proven.

6) Logically, when one performs perturbative matching of the fundamental theory on the effective theory, one does it for observables which are computable in both theories, and at an energy scale where both theories are perturbative. If this condition is not satisfied the matching does not deserve to be called perturbative. The transition matrix that the authors propose to match does not seem to satisfy these requirements. The transition matrix itself does not seem to have a good perturbative expansion nor in fundamental nor in the truncated theory. It's only when one subtracts the two large terms one gets a small correction term. It would be nice to have a discussion to why this is allowed, and perhaps include references to other similar uses of Effective Field Theory, not fully legitimate at first appearance but giving correct results.

7) The status of the condition E_i,E_f<<E_max in the matching correction calculations is not fully clear. They use this condition to simplify those calculations since in this limit a local approximation can be derived. Note that if the theory is really perturbative at E_max, then by using exact (not approximated) correction terms for all the states up to E_max the results should only improve. This would be an important check on their scheme. For the previous scheme of Rychkov and Vitale such a check, done by Elias-Miro, Montull, Riembeau http://arxiv.org/abs/1512.05746, showed that in fact full correction terms gave worse results than approximate local ones. The troublesome states for which perturbativity was violated were identified; they corresponded to states with many particles. If the authors have an opinion on whether the same issue would affect their scheme, it would be nice to add it. (See the discussion below Eq. (2.14) in Elias Miro, Rychkov, Vitale http://arxiv.org/abs/1706.09929)

8) (Related to comment 1) It would be nice to comment why correction terms computed under the condition E_i,E_f<<E_max are subsequently used for all the states up to the energy E_max in the subsequent numerical computation. According to Eq. (6.7) with its H_0/E_max dependence of C_na, correct matching corrections for E_i,E_f~E_max may differ at order 1 from the ones they used.

9) Concerning the use of Euler-Maclaurin formula: k_min in (6.4) is given by the integer part of the r.h.s. of (6.3). This means that the l.h.s. of (6.5) has some step-discontinuities in E_max. There are limitations in approximating such step-discontinuous functions with a series in inverse powers. I wonder if the authors have any comment on this and how it is related to the "scatter" in their numerical plots?

---

## Round 1 · Referee Report · Anonymous (Referee 1) · 2022-1-7

Report

The authors introduce a new strategy for performing Hamiltonian Truncation calculations. The idea consists in constructing the truncated Hamiltonian Heff by matching a transition amplitude T. Heff is uniquely determined by requiring that the computation of the transition amplitude T in the finite low energy Hilbert space equals the computation of T in the fundamental theory. The authors test this idea with phi4 theory in two dimensions, and numerically obtain a nice looking spectrum at strong coupling.

This work opens a new pathway for Hamiltonian Truncation calculations. It is a very interesting paper with original results and novel analyses. Therefore I will recommend its publication after the authors address the observations below, and clarifications I would like to ask:

1) The Heff introduced in this paper is non-hermitian simply because the authors are matching a non-hermitian operator. Hermitian matrices have nice properties of reality of the eigenvalues and orthogonality of the eigenstates. Therefore I believe it would be nicer to work with Hermitian effective Hamiltonian. This could be fixed by matching a hermitian version of the author’s T operator, i.e. 1/2(<f|T |i > + h.c.).

Interestingly, matching the operator 1/2(<f|T |i > + h.c.) leads to a Heff which coincides with the so called Schrieffer-Wolff (SW) effective Hamiltonian. The SW is a correct effective Hamiltonian by construction, because it is obtained by performing a canonical transformation to block-diagonalize the fundamental Hamiltonian (this construction is applied to a UV renormalized fundamental theory). Therefore the SW derivation justifies the procedure of matching the operator 1/2(<f|T |i > + h.c.) as a route for arriving to the SW effective Hamiltonian. This reasoning leads one to ask whether the Heff of the authors (retrieved by matching <f|T |i > instead of 1/2<f|T |i > + h.c. ) is exactly correct, or only approximately? What I am pointing out is a matter of principle, whose consequences may not be detectable by computing the spectrum of the two dimensional phi4 theory with second order Heff.

2) Equations 2.24 are the correct solution to the matching equation at O(V^3). However, towards the end of section 2, the authors claim that the correct matching equations require the extra step of expanding equations 2.24 in powers of 1/Emax. Should one keep only the leading 1/Emax correction in 2.24, or keeping the leading and subleading is also fine? Is one allowed to incorporate the whole series in 1/Emax, and if so, why expanding in powers of 1/Emax in the first place? The justification “in accordance with standard effective field theory methodology” is unclear because the problem at hand departs from standard quantum field theory in many interesting ways.

3) The authors also show that their calculations obey various expectations of EFT regarding separations of scales. In particular, a definition is given in the introduction: the coefficients of the Heff should be dominated by states close to the cutoff. This is so because Heff is obtained from a renormalized (i.e. finite) fundamental theory. Fine. However the authors take this expectation one step further in the main text, and show that the separation of scales to O(V^2) is obeyed in a local manner. Namely, it turns out that leading UV/IR overlaping regions in momentum space (of the coefficients of the local, un-normal ordered, operators in Heff) vanish. This is a manifestation of the locality of Heff at this order of the calculation: states containing soft momentum modes can have energy close to the cutoff, however these states do not dominate the contribution to the coefficients of Heff.

4) Section 6 contains a power-counting discussion. Reading through it, one has the impression that non-locality and non-hermicity are somewhat intertwined. However, as I emphasised above this does not need to be the case.

5) The authors carefully estimate the error of their approximation in Heff at second order. This allows them to show that raw spectrum converges as 1/Emax^2 while leading order second order as 1/Emax^3. The focus of the numerical analysis is on convergence. However, with such a good control on the convergence, it would had been interesting to compute universal data at the critical point. Those computations could be compared across different methods and works.

6) Equation 8.3 is wrong. There the authors are referring to the exact eigenvalue equation 2.3, therefore the denominator of the exact “Delta H_exact” should be Epsilon-H0-V instead of Epsilon-H0. References [5,6] did two essential approximations: 1) substituted the denominator of Delta H_exact by Epsilon-H0 ; 2) approximated V.1/(Epsilon-H0).V by local operators.

---

## Round 1 · Referee Report · Balt van Rees (Referee 3) · 2022-1-14

Report

The paper offers a new view on improving the Hamiltonian truncation method.

The authors propose that the effective Hamiltonian, as a matrix, must reproduce an observable 'related to the S-matrix'. They propose to compute this observable perturbatively with the real Hamiltonian and use this to determine the effective Hamiltonian.

I think the matching of an entire matrix of observables is an interesting proposition. (The choice of the off-diagonal terms in the counterterms was also discussed in section 4.2 of 1803.05798, where greater emphasis was put on Hermiticity.) The authors however do not explain why their particular observable is the right one to match? Would there not be a better one, perhaps one that avoids a non-Hermitian Hamiltonian?

The paper then proceeds by computing, in the example of phi^4 theory in two dimensions, the effective Hamiltonian so defined to second order in perturbation theory. The authors choose to then restrict themselves to a so-called 'local approximation' where only coupling renormalizations remain.

In numerical experiments it is observed that the cutoff dependence of the estimations is improved from E_max^{-2} to E_max^{-3}. The authors claim this is 'a strong indication that our method is working as expected'.

I am puzzled by such a strong claim. The local approximation to leading order in E_max seems to me entirely equivalent to the improvement computable by standard perturbation theory. What part of the numerical results relies on matching an entire matrix of observables rather than just the energies themselves?

The paper is in my opinion clearly written; my only stylistic comment would be that a punchline equation is sometimes missing. In particular, the effective Hamiltonian 'described above' (on page 30) needs to be cobbled together by the reader from several equations scattered throughout the previous section.

Requested changes

I would like to see the above questions answered.

---

## Round 2 · Referee Report · Anonymous · 2022-4-29

Report
The authors have addressed thoroughly all the points I raised (referee 1),
and have provided detailed account of the modifications done to the preprint.
I therefore recommend the publication of this interesting paper in SciPost.

---

## Round 2 · Referee Report · Balt van Rees · 2022-5-12

Report
I think the paper has improved substantially and is suitable for publication in SciPost physics. I am grateful to the authors for addressing my comments.

---

## Round 2 · Referee Report · Slava Rychkov · 2022-5-18

Report
I thank the authors for the thorough revision. I was happy to see the new equations 7.6a, 7.6b which are crucial for the reproducibility of their results by the others.
1) I was disappointed that they made no attempts to extablish the parametric behavior of 7.6a, 7.6b with E_max, m, R. The comment after (5.26) says that these are dominated by contribution of states near the cutoff. For large R/m, and large Emax, the spectrum near the cutoff is dense, so these expressions should have an analytic expression at least in this limit, which would be interesting to see. I hope they can do this in the future work, so that they compare their effective Hamiltonian to that by Rychkov and Vitale (see my previous report). The computation should be relatively easy following the techniques by Elias-Miro, Rychkov and Vitale https://arxiv.org/abs/1706.09929, Appendix F.2.
2) I am not convinced that one is always allowed to match using a quantity which gets contributions from a strongly coupled region. How can we be sure that all things from the strongly coupled region which were supposed to cancel, canceled correctly without leaving a small finite error, if these contributions were not computable in the first place, being strongly coupled? This needs a separate model-depending check.
I would like to mention here as an example my old work https://arxiv.org/abs/1211.5543, Section 3.4, where matching was done in a range of energies where both effective and matched theory could be trusted, and where the matched quantity could be computed in both theories (without getting contributions from a strongly coupled region). This shows that matching can be done safely, if one wishes so.
In spite of these remaining misgivings I approve this paper for publications, hoping that these issues will continue to be clarified in future work.

---

## Round 2 · Author Response

We thank the referees for the exceptionally detailed and helpful feedback. We apologize for the delayed response, but it took us some time to digest and properly respond to all of the comments and suggestions in the referee report. We have made extensive revisions to the paper in a good-faith effort to incorporate the insights of the referees and address their comments. We hope that the present form of the paper is acceptable for publication.
Below we provide responses to the referee comments point by point, using the same numbering as in the referee reports. A complete list of changes in the paper is listed separately at the end.
Sincerely,
Timothy Cohen, Kara Farnsworth, Rachel Houtz, Markus A. Luty

---

## Round 2 · List of Changes

Referee 1:
1. All three referees commented that non-Hermiticity may not be a fundamental feature of the effective Hamiltonian. We will therefore address the comments of all three referees on this question here.
First, we would like to emphasize that we have defined the effective Hamiltonian by matching. This definition is then used as the basis for a systematic expansion in powers of 1/Emax. For example, it is not the same as the `exact Hamiltonian' used in previous references, which depends on the eigenvalue that is being computed. There may well be other definitions that also work, possibly defining a Hermitian effective Hamiltonian with the same spectrum. We think this is an interesting question, and we do not know what is possible. In this paper, we have focused on investigating the definition that we have proposed. The calculational and numerical checks that we have performed work as expected, and encourage us to continue to pursue this direction, which we plan to do in future work.
To address the concerns of the referees regarding the non-Hermiticity of the effective Hamiltonian, we have removed statements throughout the paper that suggested that non-Hermiticity is a fundamental feature of the effective Hamiltonian. We have also added a very preliminary discussion of some alternative definitions that may give a Hermitian effective Hamiltonian on pp. 11-12 and appendix B.
Referee 1 makes the proposal that one can define a Hermitian effective Hamiltonian by matching the Hermitian operator T +T^\dagger, rather than T, as we do in this work. This is an interesting suggestion, but unfortunately it does not work. We checked that at O(V^3) the effective Hamiltonian defined this way is ill-defined due to IR divergences. This is a manifestation of the failure of separation of scales. We have added a comment
on this to the discussion of non-Hermiticity on pp. 11-12.
Referee 1 states that matching T +T^\dagger gives the Schrieffer-Wolff effective Hamiltonian. We feel that the connection to the Schrieffer-Wolff effective Hamiltonian is rather distant from the main topic of the paper, and we have added only a brief mention of it in the paper. Briefly, here is our understanding of this relation. The result of performing the Schrieffer-Wolff similarity transform gives the O(V^2) contribution to the effective Hamiltonian
<f|H_2|i> = 1/2 \Sum_\alpha <f|V|\alpha><\alpha|V|i>(1/E_{i\alpha} + 1/E_{f\alpha}). (1)
Note that the sum is over all states $\alpha$. For this to be a useful effective Hamiltonian, one assumes that the low-energy states are separated by a large gap \Delta E from the high-energy states, and the unperturbed Hamiltonian H_0 is diagonalized on the low-lying states. With these assumptions, the sum only gets contributions over the high energy states $\alpha$, and the effective Hamiltonian gives an expansion in powers of 1/(\Delta E). Matching T +T^\dagger at O(V^2) gives the same equation as Eq. (1), but the sum over $\alpha$ is over high energy states whether or not H_0 is diagonal in the low-energy subspace. The 1/(\Delta E) expansion of the Schrieffer-Wolff effective Hamiltonian is very different from the 1/Emax expansion we are performing. We do not feel that explaining this in detail is appropriate for this paper, and we hope the referee will agree that the changes we have already made in the paper are adequate.
Referee 2 points out that we may be able to define a Hermitian effective Hamiltonian using a similarity transformation. We thank the referee for this suggestion, which we believe is interesting to pursue. We have commented on this in the main text, and discussed a very preliminary attempt to find such a similarity transformation in appendix B.
Referees 1 and 2 also point out that in the `local approximation' considered so far, the non-Hermiticity (and non-locality) do not appear, and therefore these aspects of our effective Hamiltonian are not tested by the calculations we have performed. We agree, and this was stated in the original version of this paper. We have added several additional statements to emphasize this point further. We are currently performing higher-order calculations where these features make their appearance, and we plan to publish them in a follow-up paper. We believe that the present results are sufficient for publication, since we have demonstrated 1/Emax^3 numerical errors from the first correction term in a parameter-free systematic expansion.
2. Referee 1 points out a lack of clarity regarding general power counting arguments in section 2. In section 2, we are considering a perturbative expansion in powers of V . This expansion is later used as the basis for the expansion in powers of 1/Emax in section 6, but this expansion depends on the model. We have attempted to make this two-step logic clearer by modifying both sections 2 and 6.
3. Referee 1 states that separation of scales is expected in the `local approximation' used in the text. We agree with the referee that it is essential to check separation of scales beyond the local approximation, and we are planning to do so in follow-up work. We have made a number of changes throughout the text to emphasize that the formalism needs to be tested beyond the local approximation. However, we would like to push back a bit on the idea that separation of scales is automatic in the local approximation. Matching T +T^\dagger (as suggested by the referee) gives IR divergences that result from vanishing energy denominators (Eq. (2.25) in
footnote 4 in the paper), and this is a violation of separation of scales that would be present even in the local approximation at O(V^3).
4. Referee 1 points out that non-locality and non-Hermiticity need not be connected in our effective Hamiltonian. We agree, and have added clarification of this point at the end of section 6.1. As already discussed in point 1 above, we have also de-emphasized the non-Hermiticity of our effective Hamiltonian throughout the paper.
5. Referee 1 suggests that we compute physical quantities such as critical exponents at the critical point. The focus of this paper is on the convergence of the 1/Emax expansion, while an accurate extraction of critical exponents requires a careful treatment of finite volume effects along with the truncation error. We request that we be allowed to leave this for future work.
6. Referee 1 points out that our Eq. (8.3) is incorrect. We thank the referee for catching this, and we have corrected it.
Referee 2:
Referee 2 states that their comments are not required to be addressed for publication, but we have made a number of changes to the paper in response to their comments.
1. Referee 2 requests extra details clarifying the exact procedure used in the numerics. We have added Eqs. (7.5)-(7.6) and the surrounding discussion to summarize our final results and explicitly present the Hamiltonian matrix we diagonalized.
Referee 2 also asks about the noise in our plots. We have added additional clarifying remarks near Eq. (5.26) to the effect that the Euler-Maclaurin approximation was not used for numerical results, and therefore does not explain this noise. We do not currently understand the origin of this noise. It is not large enough to be important for the results in this paper, although it will be important when we go to next order in the expansion, and we will address it then.
2. Referee 2 suggests that we clarify the relationship between our method and his previous work, which also make use of the local approximation, but does not obtain 1/Emax^3 errors. We agree that this would be interesting, but we believe it is more important for us to understand our method better by going to higher orders in our expansion. We do not know how to give a simple formula for the correction as in the referee's
work because the contribution from the states near the cutoff appears to be quite complicated, and not amenable to a simple approximation.
3. See point 1 for Referee 1.
4. Referee 2 asks about the physical meaning of the transition matrix T. This is a great question, and we do not have a good answer. Since we only use it for matching, it does not even have to have a physical meaning, it just has to define a good effective Hamiltonian. We added a comment on p. 9 pointing out that T suffers from an IR
divergence even in finite volume. This IR divergence cancels in the matching (separation of scales again), so this does not invalidate our method, but it may explain why T by itself is not useful.
5. The referee suggests we explicitly say that the diagonal elements of Eq. (2.23) agree with what is obtained by matching the spectrum. We have added this in the paragraph after Eq. (2.23). (The fact that this was not included in the original draft was an oversight on our part, we certainly did perform this crucial check!)
6. Referee 2 states that while the matching contributions are small, the contributions to T from the low-energy and high-energy theories are large. This is because T gets contributions from all scales, and the cancelation comes because the strongly-coupled IR region cancels in the matching. This cancelation is a manifestation of the separation of scales that is an essential (but standard) feature of EFT matching. We have added references to general effective field theory methodology where this is explained.
7. The referee asks whether we have considered how our results would change if we attempted to match the effective Hamiltonian for states near the cutoff. We have not done so. The standard EFT methodology is to expand all quantities around zero momentum and energy, corresponding to a derivative expansion of the effective theory. The fact that this works as expected in the calculations we have performed so far
motivates us to continue in this direction.
8. See point 1 above.
9. See point 1 above.
Referee 3:
Referee 3 did not provide numbered points, but asked us to address 3 questions (paraphrasing): Why is T the right quantity to match? Is there another quantity that could be matched to define a Hermitian effective Hamiltonian? Do the numerical results depend on the precise definition of the matching?
We believe that we have addressed all these questions with the changes discussed in point 1 of Referee 1.
Referee 3 also suggests that we more clearly summarize our main results. We have addressed this in point 1 of Referee 2.
List of changes to the draft:
Since we made many changes, we felt it would be helpful to provide a complete list of the substantive ones here to make it easy for the referees and editor to follow them.
p. 1: Changed wording in the abstract.
p. 5: Modified the bullet on the non-Hermiticity
p. 5: Added a bullet emphasizing the 1/Emax expansion
p. 6: Added a mention of the new Appendix B
p. 6: Added footnote 2
pp. 8-9: Added clarifying discussion to the beginning of Section 2.2
pp. 11-12: Added a few paragraphs at the end of Section 2.3 to explain what is in the new Appendix B
p. 20: Added clarifying remarks to the end of the second paragraph at the beginning of Section 5
p. 23: Added clarifying remarks above Eq (5.12)
p. 24: Added clarifying comments around Eq (5.19)
p. 26: Added clarifying discussion below Eq (5.26)
pp. 26-27: Expanded the first paragraph of Section 5.5
p. 27: Shortened what is now footnote 10
p. 30: Removed the last paragraph before Section 6.1
p. 30: Heavily edited the first paragraph of Section 6.1
p. 31: Added a discussion of non-locality in the paragraph below Eq (6.6)
p. 34: Added clarifying equations to the beginning of Section 7.2, Eqs (7.5) and (7.6)
p. 35: Added a reference to the python numerical package `scipy.sparse.linalg'
p. 42: Added Eq (8.3) and related comments right below it
pp. 50-51: Added Appendix B on similarity transformations to make the effective Hamiltonian Hermitian

---

## Editorial Decision

published